# Optimized Covariance Design for AB Test on Social Network under Interference

**Qianyi Chen**[1], **Bo Li**[1]*, **Lu Deng**[2], **Yong Wang** [2]
[1]School of Economics and Management, Tsinghua University, China
[2]Tencent Inc., Shenzhen, China
`cqy22@mails.tsinghua.edu.cn, libo@sem.tsinghua.edu.cn`
`adamdeng@tencent.com, darwinwang@tencent.com`

## Abstract

Online A/B tests have become increasingly popular and important for social platforms. However, accurately estimating the global average treatment effect (GATE) has proven to be challenging due to network interference, which violates the Stable Unit Treatment Value Assumption (SUTVA) and poses great challenge to experimental design. Existing network experimental design research was mostly based on the unbiased Horvitz-Thompson (HT) estimator with substantial data trimming to ensure unbiasedness at the price of high resultant estimation variance. In this paper, we strive to balance the bias and variance in designing randomized network experiments. Under a potential outcome model with 1-hop interference, we derive the bias and variance of the standard HT estimator and reveal their relation to the network topological structure and the covariance of the treatment assignment vector. We then propose to formulate the experimental design problem as to optimize the covariance matrix of the treatment assignment vector to achieve the bias and variance balance by minimizing a well-crafted upper bound of the mean squared error (MSE) of the estimator, which allows us to decouple the unknown interference effect component and the experimental design component. An efficient projected gradient descent algorithm is presented to the implement of the desired randomization scheme. Finally, we carry out extensive simulation studies[2] to demonstrate the advantages of our proposed method over other existing methods in many settings, with different levels of model misspecification.

## 1 Introduction

A/B test, also known as controlled experiment, is an effective way to evaluate treatment effect of new product features and facilitate data-driven decision-making [14]. Due to its simplicity and efficiency, it has become the gold standard in the industry. The theory of A/B test is based on the Stable Unit Treatment Value Assumption (SUTVA), under Neyman-Rubin's framework [33], which assumes unit's treatment effect isn't influenced by other units' treatment assignments. However, in situations where social network connects units, SUTVA is often violated since an individual's behavior can be influenced by their friends. This violation of SUTVA is referred as network interference or social interference, which brings challenges to both estimation and inference of parameters of interest. It has been shown that ignoring network effect can induce much higher bias or variance [29].

To tackle the bias and variance induced by network interference, researchers have tried to design randomization schemes in pre-treatment phase and estimators or adjustment methods in post-treatment phase. In this paper, we will focus on designing randomization scheme, namely, experiment design.

---

*Corresponding author
[2]The code is available at `https://github.com/Cqyiiii/Optimized_Covariance_Design-NIPS2023`

37th Conference on Neural Information Processing Systems (NeurIPS 2023).

Specifically, we take the graph structure as known and leverage it to assist experiment design. Our basic scheme is cluster-level randomization, which treats cluster or community in the graph as unit of treatment and all units in the same cluster share the same treatment. [19] is an early work investigating this approach, and [35] proposes to form clusters utilizing graph. The graph cluster randomization is conceptually simple but effective in many scenes, and thus becomes a prevalent paradigm for network experiment design. Moreover, we consider experimental design on a social network in this work. Social network typically has organic structure and high clustering coefficient, which corresponds to meaningful cluster structure and also motivates us to adopt cluster-level randomization. We assume the cluster structure of the network is given. In practice, interpretable community detection algorithms such as the Louvain algorithm [4] are typically applied for cluster construction. Moreover, we stress the scalability of our method since social network is usually large-scale in reality.

Related works have shown that there exists an explicit bias-variance tradeoff [23, 41]. However, many HT-style estimators that utilize exposure indicator to ensure unbiasedness are designed on the basis of the neighborhood interference assumption that restricts global interference to 1-hop neighborhood [10, 25, 36, 39, 9], and graph-dependent generalized propensity score [1]. Nonetheless, we argue that pursuing unbiasedness by simple sample trimming leads to small effective sample size with high variance, and thus is far from optimal. Moreover, computational cost is also high in this approach since a large number of Monte Carlo simulations are usually needed for computing the generalized propensity score, or the trimming probability for constructing sample weights. Having observed the preceding drawbacks of the unbiased estimator, we consider the standard unit-level HT estimator to achieve possible better bias and variance tradeoff with less computation burden at the meantime. Based on a potential outcome model, we characterize the relationship among the bias, variance, the network topology, and the treatment assignment mechanism. This development allows us to design experimental schemes that minimizes MSE and achieve better bias-variance tradeoff.

Besides from the time-consuming Monte Carlo simulation for calculating generalized propensity score mentioned above, the computational efficiency of experiment design is indeed a limitation of current methods. For instance, the two-wave experiment [37], entails solving a mixed-integer quadratic programming problem, which can be computationally expensive. The adaptive covariates-balancing design [25] adopts a sequential process for treatment assignment and thus isn't very scalable. To address this limitation, we propose to utilize gradient-based optimization algorithm.

In this paper, we propose to treat the covariance matrix of treatment vector as decision variable in optimization, which is nontrivial as a result of interference and corresponding complex variance of estimator. Similar approach in optimization step is proposed in [17], where interference isn't considered, and making covariance covariance acting as decision variable is a natural choice owing to the simple structure of variance there. We draw attention to the fact that related works concerning experiment design mostly concentrates on a predetermined class of randomization schemes and seeks to optimize the parameters to obtain designs that minimize variance or MSE of estimator [37, 23, 5, 42]. However, we argue that these designs can all be understood as different patterns of the covariance matrix of the treatment vector, and we propose to treat the covariance matrix as decision variables rather than limiting our scope to a particular class of designs. One of major challenges with this approach is generating a randomization scheme that supports legitimate sampling, as there may not be a joint distribution that corresponds to the optimized covariance matrix in our formulation. To address this issue, we propose to adopt projected gradient descent to guarantee the validity of optimized covariance matrix. Furthermore, we also compare our method with other popular designs through extensive simulation on a real online social network. Various interference intensity levels and potential outcome models are considered. It is found that our method is robust to model misspecification and outperforms in almost all settings, especially when the interference isn't too weak compared to the direct treatment effect, which is the scene we target.

**Contributions.** Our contributions are summarized as follows.

1. We derive a well-crafted upper bound for the MSE of the HT estimator that decouples the estimation of causal mechanism and experimental design. This enables us to optimize the experimental design by minimizing this bound, in which the covariance matrix of treatment vector acts as decision variables.

2. We propose the formulation of covariance optimization problem with reparameterization of covariance matrix and constraints that guarantees legitimate sampling subject to optimized

covariance matrix, and a projected gradient descent algorithm is proposed for solving the optimization problem.

3. We conduct systematic simulation on a social network and compare our method with several methods proposed recently under a range of settings, providing credible reference for the effectiveness of our method.

**Related Works.** A variety of interference has been considered in related works[12], such as spatial [23], temporal [13] and network, which is the scene we target. Besides from the general network, there is also another line of work focused on a two-sided marketplace or platform, where the graph is bipartite [43, 24, 16, 30, 27].

In the field of network experiment design, most of extant research assumes that the graph structure is known and leverage it to assist in modeling the interference[2], though several graph agnostic methods are developed these days [40, 8]. Nevertheless, the interference is conducted through edges of network, and agnostic methods aren't effective enough.

Beyond unit-level randomization, cluster-level randomization on graph has attracted extensive attention, including both experimental design [23, 25, 5, 30] and estimation [26, 1, 6, 15]. The seminal paper of this line of work is [35], which proposes graph cluster randomization that treats clusters as units in randomization. This methodology is motivated by providing a better lower bound of exposure probability in HT estimator that uses exposure probability as inverse weight, aiming at controlling the variance of such an unbiased estimator. Along the same direction, [36] proposes a polynomial lower bound on exposure probability through randomized local clustering algorithm.

Irregular global interference may render the experimental design problem intractable [39], thus structural assumptions are usually introduced to rule out such arbitrary interference. The first category is partial interference [34, 19, 3, 11, 5], which assumes disjoint community structures and interference only happens within each community. The second category is neighborhood interference, which assumes the interference solely originates from units' 1-hop neighborhood [25, 36, 41, 12, 8]. This kind of assumption is more practical compared with partial interference, and thus widely applied. The third category is potential outcome model, which directly assumes the expression of potential outcome, e.g. classical Cliff-Ord auto-regressive model [7], linear potential outcome models[25, 39]. Actually, some potential outcome model implies neighborhood interference assumption in its mathematical expression. It's also worth to be noted that there also appear works trying to weaken the structural assumptions mentioned above and consider the interference of more general form [6, 22, 23, 18, 15].

## 2 Basic Framework

### 2.1 Setup

We consider a finite population of $n$ units, and the treatment vector is represented as $\boldsymbol{z} = (z_1, z_2, \ldots, z_n) \in \{0, 1\}^n$. In this paper, we consider the experiment design problem in which a sensible random treatment assignment mechanism of all units is designed to achieve good performance of the resultant treatment effect estimation. We follow the Neyman-Rubin's framework [33]. Without SUTVA assumption, we define the potential outcome of unit $i$ as $Y_i = Y_i(\boldsymbol{z})$, which means the potential outcome of unit $i$ can depend on the treatment assignment of all units.

The parameter of interest is the global average treatment effect (GATE), which is denoted by $\tau$,

$$\tau := \frac{1}{n} \sum_{i \in [n]} \left( Y_i(\mathbf{1}) - Y_i(\mathbf{0}) \right) \tag{1}$$

Here we use $[n]$ to denote set $\{1, 2, ..., n\}$, and $\mathbf{1}(\mathbf{0})$ is the $n$-dimensional vector of $1(0)$, which corresponds to global treatment and global control.

We denote the social network as $\mathcal{G} = (\mathcal{V}, \mathcal{E})$. The node set $\mathcal{V}$ is $[n]$, corresponding to all units considered above. Edge set $\mathcal{E}$ represents the social connection between units. Furthermore, we use $d_i$ to denote the degree of node $i$. We assume the graph structure $\mathcal{G}$ is known in advance.

In this paper, we consider graph cluster randomization [35]. Here we also assume that cluster structure is given, which is generated from certain community detection algorithm. We define $S_k$ as the $k$-th

cluster, where $k \in [K]$. Clusters compose a splitting of node set, namely,

$$S_k \subset [n] \quad S_i \cap S_j = \emptyset \quad \cup_{k=1}^{K} S_k = [n] \tag{2}$$

The graph cluster randomization implies that units in the same cluster share the same treatment assignment, namely,

$$z_i = z_j \quad \forall i, j \in S_k, k \in [K] \tag{3}$$

Since we consider randomization scheme at cluster level, it's more convenient to define cluster-level treatment vector $\boldsymbol{t} = (t_1, t_2, ..., t_K) \in \{0, 1\}^K$. Specifically, we consider the balanced cluster-level randomization scheme satisfying

$$\mathbb{P}(t_k = 1 | \mathcal{G}) = \frac{1}{2} \quad \mathbb{E}[z_i] = \frac{1}{2} \tag{4}$$

## 2.2 Potential Outcome Model and Estimators

One of the key ideas of this paper is to deduce an appropriate objective function to optimize experiment design schemes under consideration. In this part, we will consider standard HT estimator, namely,

$$\hat{\tau} = \frac{1}{n} \sum_{i \in [n]} \left( \left( \frac{z_i}{\mathbb{E}[z_i]} - \frac{(1 - z_i)}{\mathbb{E}[1 - z_i]} \right) Y_i(\boldsymbol{z}) \right) \tag{5}$$

Then we introduce the considered potential outcome model.

**Assumption 1 (Potential Outcome Model)** *The potential outcome of unit $i$ is generated by*

$$Y_i(\boldsymbol{z}) = \alpha_i + \beta_i z_i + \gamma \sum_{j \in N_i} z_j \tag{6}$$

*where $N_i$ is the 1-hop neighborhood of unit $i$.*

In this potential outcome model, $\beta_i$ can be interpreted as direct treatment effect of unit $i$, and $\gamma$ measures the interference effect, characterizing the extent to which each unit is affected by its neighborhood in the network. This model is a simplified (in interference part) version of over-parameterized model in [39], in which heterogeneous interference effects are accommodated. We make a somewhat stronger assumption to simplify the structure of the bias, variance and MSE. Under this model, we can express the GATE in terms of the potential outcome model parameters,

$$\tau = \frac{1}{n} \sum_{i \in [n]} (Y_i(\boldsymbol{1}) - Y_i(\boldsymbol{0})) = \frac{1}{n} \sum_{i \in [n]} (\beta_i + \gamma d_i) \tag{7}$$

We will then derive the bias and variance of $\hat{\tau}$ under this potential outcome model. We point out that the number of unknown parameters in this model is still greater than the number of known potential outcomes, which can be understood as overparameterization, and we only utilize it to derive an optimizable MSE upper bound that's not concerned with estimation of $\beta_i$ and $\gamma$ in this model. In fact, when the interference effect and the direct treatment effect are comparable at cluster-level, a multiplier of the MSE upper bound can be factored out, which is crucial in our methodology and will be shown in section 3.1.

## 2.3 Bias of HT Estimator

Based on the potential outcome model, we can derive the bias and variance of our estimator explicitly. We firstly introduce cluster-level treatment assignment, $\boldsymbol{t} = (t_1, t_2, ..., t_K)$, and matrix $\boldsymbol{C}$ summarizing edges between(within) clusters with its elements defined as

$$\boldsymbol{C}_{ij} = |\{(u, v) : (u, v) \in \mathcal{E}, u \in S_i, v \in S_j\}| \tag{8}$$

We have following proposition about bias.

**Proposition 1 (Bias of HT Estimator)** *The bias of standard HT estimator is*

$$\mathbb{E}[\hat{\tau}] - \tau = \frac{\gamma}{n} \left( 4 \operatorname{trace}(\boldsymbol{C} \operatorname{Cov}[\boldsymbol{t}]) - \sum_{i,j \in [K]} \boldsymbol{C}_{ij} \right) \tag{9}$$

*where* $\operatorname{Cov}[\boldsymbol{t}]$ *refers to the covariance matrix of treatment vector* $\boldsymbol{t}$.

Specifically, we can delineate the source of bias in our setting. If we consider graph cluster randomization as basic scheme, edges within cluster don't contribute any bias, while every edge between clusters contributes to bias according to the covariance of cluster-level treatment assignments. This enlightens us that independent treatment assignments is far from optimal, and we can reduce bias by assigning treatments with appropriate positive correlations that should be higher for more densely connected cluster pair. Furthermore, we point out that bias may be of no account when a very sparse network is considered, but it indeed counts in our context of social network.

We also note that the treatment vector is summarized by its covariance matrix in the expression of bias. Hence, we denote the bias by $B(\operatorname{Cov}[\boldsymbol{t}])$.

## 2.4 Variance of HT Estimator

To derive the variance, we plug in the potential outcome model and express the estimator as

$$\hat{\tau} = \frac{2}{n} \sum_{i=1}^{n} -\alpha_i + (\beta_i + 2\alpha_i - \gamma d_i)z_i + 2\gamma \sum_{j \in N_i} z_i z_j \tag{10}$$

An important difference in the derivation of variance contrasting that of bias is all base levels $\alpha_i$ is no longer negligible. Hence, we consider to target at the scenes that we can acquire knowledge of all base levels in advance.

**Assumption 2 (Known Base Levels)** *The base level* $\alpha_i$ *of unit* $i$ *is known in advance for all units* $i$.

This assumption is strong in the classic context of experiment design, while we propose to target the scene where AB tests are launched by social platforms. On the one hand, the improvement in user activity indices brought by new traits or interventions, specifically, direct treatment effect and interference, is typically much smaller than the base levels $\alpha_i$, which means the experiment design may be quite challenging when agnostic to base levels. On the other hand, it's very reasonable to assume that we know all $\alpha_i$s before conducting experiments since those indices we're concerned with are recorded and/or estimated daily or weekly by social platforms.

Utilizing this assumption, we can first adjust our estimator as

$$\hat{\tau}_{-\alpha} = \frac{1}{n} \sum_{i \in [n]} \left( (\frac{z_i}{\mathbb{E}[z_i]} - \frac{(1 - z_i)}{\mathbb{E}[1 - z_i]})(Y_i(\boldsymbol{z}) - \alpha_i) \right) \tag{11}$$

This adjustment has also been recommended by [39]. Notice that this adjustment helps us remove the influence of base level $\alpha_i$ when we consider variance. In the following, we'll adhere to this format and proceed to derive the variance term, which is summarized in the following proposition.

**Proposition 2 (Variance of HT Estimator)** *The variance of standard HT estimator is*

$$\operatorname{Var}[\hat{\tau}] = \frac{4}{n^2} (\operatorname{trace}(\boldsymbol{h}\boldsymbol{h}^T \operatorname{Cov}[\boldsymbol{t}]) + 4\gamma \operatorname{Cov}[\boldsymbol{h}^T \boldsymbol{t}, \boldsymbol{t}^T \boldsymbol{C} \boldsymbol{t}] + 4\gamma^2 \operatorname{Var}[\boldsymbol{t}^T \boldsymbol{C} \boldsymbol{t}]) \tag{12}$$

*where* $\boldsymbol{h}$ *is the vector* $(\sum_{i \in S_k} \beta_i - \gamma d_i)_{k=1}^K$.

Notice that the expression of variance contains second-order term with regard to $t$, namely, $\operatorname{Cov}[\boldsymbol{t}]$, as well as third and fourth-order terms involving $\boldsymbol{t}^T \boldsymbol{C} \boldsymbol{t}$, a quadratic form of $\boldsymbol{t}$. It's feasible to analyze the variance precisely with a specific randomization scheme, such as the independent block randomization [5], but it is generally not straightforward to optimize a broad set of randomization schemes in its exact form. In the following subsection, we will derive a bound of variance to transform third and fourth-order terms into second-order, which also motivates our optimized covariance design.

We comment that a bias-variance tradeoff is unveiled in the previous two propositions. Actually, $C$ is a matrix with non-negative elements, and introducing positive covariance between treatment assignments of different clusters will reduce the bias. On the other hand, introducing positive covariance means an increment in variance. In addition, though this paper doesn't involve upstream clustering algorithm, the resolution of clustering can also impact the relative importance of bias and variance, namely, low resolution usually corresponds to a higher proportion of edges within cluster, which weakens the importance of bias and stress on variance.

## 3  Optimized Covariance Design

### 3.1  Constructing Objective for Optimization

Although we have derived the expression of bias and variance, the expression of variance is complicated due to the existence of treatment effects $\beta_i$s, the interference effect $\gamma$, and the higher-order terms involving the treatment vector. We will bypass the first hurdle by adding reasonable assumption about the direct treatment effect and interference effect. The second challenge will be addressed by a upper bound argument.

We first consider the former issue, which is the relationship between $\beta_i$ and $\gamma_i$. Since we consider the interference, we propose to restrict our scope to the situation that network interference isn't too weak, which is formulated as following assumption.

**Assumption 3 (Comparability between Direct Treatment Effect and Interference)** *Given potential outcome model in equation (6), we assume there exists a constant $\omega > 0$ such that*

$$|\boldsymbol{h}_k| \leq \omega\gamma(\sum_{i\in S_k} d_i) \tag{13}$$

*holds for each cluster $k \in [K]$.*

This assumption actually demands the magnitude of interference is comparable to direct effect at cluster level. Namely, we allow a significant difference between direct treatment effect $\beta_i$ and interference $\gamma d_i$ for certain units, but there isn't too much difference after aggregation in cluster. The constant $\omega$ can be viewed as degree of tolerance on incomparability between direct causal effect and interference and depends on the both nature of potential outcome and graph structure. It's natural that people may be more susceptible to the states of their friends in some social metrics. Since we'll utilize the MSE bound as optimization objective, this constant can also be viewed as tuning parameter that's determined by domain knowledge or pilot experiment. To demonstrate the robustness of our method, we don't tune $\omega$ according to the result and fix $\omega = 1$ in the simulation, which corresponds to a moderate constraint on comparability. Empirically, our result isn't sensitive to the value of $\omega$.

We then discuss the latter issue. We bound all high-order terms by second or lower order terms. This effort leads to an upper bound which allows us to simply optimize the covariance matrix.

**Proposition 3 (Variance Bound)** *The variance of the standard HT estimator has following upper bound,*

$$\mathrm{Var}[\hat{\tau}] \leq \frac{8\gamma^2(\omega^2 + 4)}{n^2} \mathrm{trace}\left(\boldsymbol{d}\boldsymbol{d}^T(\mathrm{Cov}[\boldsymbol{t}] + \frac{1}{4}\boldsymbol{1}\boldsymbol{1}^T)\right) \tag{14}$$

*where $\boldsymbol{d}$ is the vector $(\sum_{i\in S_k} d_i)_{k=1}^K$.*

In brief, we choose to bound the third-order term by the sum of second and fourth-order terms, and bound the fourth-order term by the definition of binary treatment vector. The format is adjusted for our optimization formulation later, where we choose the covariance matrix as decision variables.

Actually, we can find that experiment design is summarized by the covariance matrix in this upper bound. Hence, we denote this upper bound by $\bar{V}_\omega(\mathrm{Cov}[\boldsymbol{t}])$. Notice that there is a common multiplier $\gamma^2$, which is associated with the unknown interference effect, in the squared bias and variance. This appealing feature allows us to optimize the MSE upper bound without estimating the parameter of the potential outcome model.

Combining proposition 1 and 3, we derive following upper bound for MSE.

**Theorem 1** *The MSE of the standard HT estimator has following upper bound*

$$\text{MSE} \leq B(\text{Cov}[\boldsymbol{t}])^2 + \bar{V}_\omega(\text{Cov}[\boldsymbol{t}]) \tag{15}$$

## 3.2 Optimizing Covariance

Now we can discuss our experiment design. For a class of parameterized randomization schemes, one can always analyze the covariance and express the bias and variance with parameters in randomization schemes. Based on the derivation in previous section, we argue that these randomization schemes is actually specifying the patterns of covariance matrix, e.g., the block diagonal pattern. We thus choose to directly optimize the covariance matrix of randomization scheme, which is actually considering a much larger hypothesis space.

Since our decision variable is a positive semi-definite matrix, it's intuitive to make attempts for formulating the problem as positive semi-definite programming with the MSE bound acting as objective. However, notice that there is still one square term with regard to $\text{trace}(\boldsymbol{CX})$, which appears in the $B(X)^2$. If we want to transform it into semi-definite programming (SDP), we must modify the objective further. Moreover, there are some intrinsic disadvantages of this formulation. Primarily, after we solve the optimal covariance matrix, we can't directly sample from the multivariate Bernoulli distribution that is subject to this covariance matrix. Secondly, method based on mathematical programming is usually precise but not efficient enough.

To guarantee that we can sample treatment vector subject to the optimized covariance, we first present a lemma that is equivalent to the Grothendieck's identity [28].

**Lemma 1** *Suppose that* $(X, Y)$ *follows a bivariate Gaussian distribution with correlation $r$, namely,*

$$\begin{pmatrix} X \\ Y \end{pmatrix} \sim \mathcal{N}\left( \begin{pmatrix} 0 \\ 0 \end{pmatrix}, \begin{pmatrix} 1 & r \\ r & 1 \end{pmatrix} \right) \tag{16}$$

*we have*

$$\text{Cov}[\text{sgn}(X), \text{sgn}(Y)] = \frac{2 \arcsin(r)}{\pi} \tag{17}$$

This lemma enables us to sample from bivariate Bernoulli distribution with mean $(\frac{1}{2}, \frac{1}{2})$ and any valid covariance. Specifically, once we solve out the optimal covariance matrix $X^*$, we can sample from multivariate Gaussian distribution with $\sin(2\pi X^*)$ acting as covariance matrix, here $\sin$ function is element-wise. Nevertheless, it must be noted that this transformation isn't omnipotent for $n > 2$ case, namely, $\arcsin(X)/2\pi$ can't recover all possible covariance matrix of $n$-variable multivariate Bernoulli distribution when $X \succeq 0$, since otherwise the multivariate Bernoulli distribution can be uniquely determined by its mean vector and covariance matrix, which is apparently false.

Moreover, we can't guarantee $\sin(2\pi X^*)$ to be positive semi-definite even if $X^*$ is positive semi-definite. Motivated by this, we adopt the Cholesky-based parameterization. We set our decision variable as matrix $R$, and we guarantee the validity of the covariance matrix of Gaussian distribution first, namely, we set the covariance matrix of Gaussian distribution as $RR^T$ and the covariance matrix of (Bernoulli) treatment vector is

$$X(R) = \frac{\arcsin(RR^T)}{2\pi} \tag{18}$$

Now we can propose our final formulation

$$\begin{aligned} \min_R \quad & M(R) = B(X(R))^2 + \bar{V}_\omega(X(R)) \\ \text{s.t.} \quad & (RR^T)_{i,j} \in [-1, 1] \quad \forall i \neq j, \; i, j \in [K] \\ & (RR^T)_{i,i} = 1 \qquad \forall i \in [K] \end{aligned} \tag{19}$$

This optimization problem is no longer convex with the introduction of $\arcsin$ function, for which we design a projected gradient descent algorithm. Since it's a non-convex optimization problem, adaptive optimizer, such as Adam [20] can be applied to improve the performance of the gradient descent step.

**Algorithm 1** Optimizing Covariance Matrix

---

**Input:** $\mathcal{G}, \{S_k\}_{k=1}^K, N, \omega$
 1: Initialization: $R_0, i \leftarrow 0$
 2: **repeat**
 3:    Calculate objective $M(R)$
 4:    $R \leftarrow \text{Adam}(R, \nabla_R M(R))$
 5:    Update $R$ with row normalization (2-norm)
 6:    $i \leftarrow i + 1$
 7: **until** $i > N$
**Output:** Optimized matrix $R^*$

---

We would like to draw attention to the fact that row normalization can be viewed as a projection operator to the domain in the aforementioned formulation, with the Frobenius norm of the matrix. This fact can be verified through the Cauchy-Schwarz inequality. Additionally, we actually optimize a total of $K \times K$ parameters using gradient information, which is both efficient and easy to implement.

After the optimization problem is solved, we can sample from it directly through a reparameterization of the multivariate Gaussian distribution,

$$t = \frac{1 + \text{sgn}(R^* \mathcal{N}(\mathbf{0}, I_K))}{2} \tag{20}$$

## 4   Simulation Study

### 4.1   Basic Setting

In this part, we carry out simulation based on a linear/multiplicative potential outcome model on a social network FB-Stanford3[32].[3] We consider following linear potential outcome model,

$$Y_i(\boldsymbol{z}) = \alpha + \beta \cdot z_i + c \cdot \frac{d_i}{\bar{d}} + \sigma \cdot \epsilon_i + \gamma \frac{\sum_{j \in N_i} z_j}{d_i} \tag{21}$$

and multiplicative model, which is a simplified version of that in [36], with removing a covariate.

$$Y_i(\boldsymbol{z}) = (\alpha + \sigma \cdot \epsilon_i) \cdot \frac{d_i}{\bar{d}} \cdot (1 + \beta z_i + \gamma \frac{\sum_{j \in N_i} z_j}{d_i}) \tag{22}$$

We choose to fix all parameters except for interference density, $\gamma$. Namely, we set $(\alpha, \beta, c, \sigma) = (1, 1, 0.5, 0.1)$ for both models, and set $\gamma \in \{0.5, 1, 2\}$ to construct three regimes. $\epsilon_i \sim \mathcal{N}(0, 1)$.

It should be noted that these two models correspond to different model misspecification forms, aiming to demonstrate the robustness of our method.

Besides our optimized covariance design (OCD), we implement following randomization schemes. We first consider two baselines, independent Bernoulli randomization (Ber) and complete randomization (CR), both of which are cluster-level. We also implement two adaptive schemes, rerandomized-adaptive randomization (ReAR) and pairwise-sequential randomization (PSR) [25, 31], which balance heuristic covariates adaptively and act as competitive baselines, since the average degree is considered as a covariate in these two methods and exactly appear in both two of our models, explicitly. Then we implement the independent block randomization (IBR) [5], which represents methods that optimize parameterized randomization schemes. Nevertheless, we argue that an appropriate objective for optimization is crucial, and we propose a randomization scheme based on IBR with optimizing variance in equation (12) as objective. We nominate it as IBR-p since the heuristic solution of this scheme in our optimization problem is pairing cluster with cluster size, namely, each block is of size 2. Notice that all randomization schemes satisfy equation (4).

For every randomization scheme we consider, we perform Monte Carlo simulation that repeats randomization schemes 10,000 times and calculates the sample mean and variance of estimators.

---

[3]Network topology data can be found in `https://networkrepository.com/socfb-Stanford3.php`

Table 1: The average bias, standard deviation and MSE of HT estimator under linear model

| gamma | 0.5 | | | 1.0 | | | 2.0 | | |
|---|---|---|---|---|---|---|---|---|---|
| metric method | Bias | SD | MSE | Bias | SD | MSE | Bias | SD | MSE |
| Ber | -0.293 | 0.521 | 0.358 | -0.588 | 0.584 | 0.688 | -1.178 | 0.709 | 1.893 |
| CR | -0.292 | 0.409 | 0.253 | -0.586 | 0.459 | 0.554 | -1.177 | 0.562 | 1.702 |
| ReAR | -0.393 | 0.227 | 0.206 | -0.700 | 0.251 | 0.554 | -1.317 | 0.303 | 1.829 |
| PSR | -0.295 | 0.235 | 0.143 | -0.587 | 0.264 | 0.415 | -1.179 | 0.323 | 1.496 |
| IBR | -0.298 | 0.273 | 0.164 | -0.593 | 0.308 | 0.447 | -1.181 | 0.380 | 1.541 |
| IBR-p | -0.294 | 0.232 | **0.141** | -0.596 | 0.261 | 0.423 | -1.185 | 0.318 | 1.507 |
| OCD | -0.198 | 0.411 | 0.209 | -0.388 | 0.469 | **0.371** | -0.764 | 0.585 | **0.926** |

Table 2: The average bias, standard deviation and MSE of HT estimator under multiplicative model

| gamma | 0.5 | | | 1.0 | | | 2.0 | | |
|---|---|---|---|---|---|---|---|---|---|
| metric method | Bias | SD | MSE | Bias | SD | MSE | Bias | SD | MSE |
| Ber | -0.365 | 0.348 | 0.255 | -0.736 | 0.394 | 0.698 | -1.475 | 0.493 | 2.421 |
| CR | -0.368 | 0.235 | 0.191 | -0.744 | 0.274 | 0.629 | -1.477 | 0.336 | 2.297 |
| ReAR | -0.402 | 0.178 | 0.194 | -0.809 | 0.174 | 0.685 | -1.548 | 0.226 | 2.450 |
| PSR | -0.366 | 0.134 | 0.152 | -0.738 | 0.153 | 0.569 | -1.479 | 0.192 | 2.227 |
| IBR | -0.369 | 0.155 | 0.161 | -0.737 | 0.178 | 0.576 | -1.484 | 0.221 | 2.252 |
| IBR-p | -0.368 | 0.163 | 0.163 | -0.739 | 0.185 | 0.581 | -1.482 | 0.232 | 2.252 |
| OCD | -0.258 | 0.040 | **0.069** | -0.517 | 0.050 | **0.271** | -1.034 | 0.054 | **1.073** |

Moreover, the oracular GATE are utilized to calculate bias.

$$\tau_1 = (\alpha + \beta + c \cdot \frac{\bar{d}}{\bar{d}} + \gamma) - (\alpha + c \cdot \frac{\bar{d}}{\bar{d}}) = \beta + \gamma$$
$$\tau_2 = \alpha(1 + \beta + \gamma) - \alpha = \alpha(\beta + \gamma)$$

(23)

At last, we set the clusters as given by Louvain algorithm with fixed random seed and resolution parameter as 10, which gives $K = 192$ clusters. We also consider other two resolution levels, namely, 2 and 5, corresponding to $K = 30, 95$, respectively. The outcome under these two levels is presented in the appendix.

## 4.2 Analysis of Results

From table 1 to table 2, we show the bias, standard deviation, and MSE of standard HT estimator. We also present the result under different estimator (such as difference-in-means estimator), different clustering resolution and different dataset (network topology) in the appendix due to limited space. Overall, it meets our expectation that the proposed method OCD demonstrates greater advantages over other methods when interference intensity, namely, $\gamma$, increases. We emphasize the outstanding performance of our method in the multiplicative model and give the credit to handling the bias-variance tradeoff better. We also discover that our approach OCD isn't sensitive to the resolution level of upstream clustering, which isn't often the case for other methods, according to the thorough simulation results presented in the appendix. In reality, when there are certain densely connected cluster pairings that are assumed to have merged but have not, the clustering outcome is poor and resolution level may be inappropriate. Such pair would typically receive a high positive covariance from the optimization process in OCD, which could be regarded as adaptively fixing the fault.

In addition, we highlight that the bias usually dominates in MSE in many settings, and it almost scales linearly with interference intensity, which is exactly as shown in the expression of bias. According to our theoretical derivation, we attribute the dominance of bias to the density of social network. There is a trend that all other methods perform similarly with two baselines, since bias is dominating in a denser network, while this component is usually ignored in the experiment design, on account of the introduction of HT estimator with exposure indicator. Moreover, although we fix $\omega = 1$ in presented result, we actually observe that our method is very robust to the choice of $\omega$ in related experiment.

Finally, we point out that the distribution of proportion of treated units of optimized randomization scheme is usually bimodal (e.g. peak at 45%, 55%) in our randomization scheme, with the mean still 50%. This bimodal pattern is the outcome of introducing non-zero covariance on almost all off-diagonal positions of covariance matrix. This is different from other methods we considered here, for which the proportion is usually strictly 50% or unimodal and concentrates at its mean, 50%.

## 5    Discussion

In this article, we derive the bias and variance of the standard HT estimator under a parametric potential outcome model and construct an optimizable objective based on the derived MSE. We then propose an appropriate formulation to transform the experiment design into an optimization problem with the covariance matrix of treatment vector acting as decision variables. Besides, we propose an algorithm based on projected gradient descent to provide optimized covariance design that supports legitimate sampling. Finally, we implement a series of methods and compare their performance in diverse settings, where the advantage of our methods is demonstrated.

We stress the significance of bias and propose our method to make attempts for balancing bias and variance better. Actually, almost all potential outcome models used in the simulation part of related works admit neighborhood interference assumption, which makes the estimator using exposure indicator unbiased or nearly unbiased. However, we argue that this assumption is reasonable but also restricts our scope. There have recently been some works considering interference beyond this assumption, where bias is also demonstrated to be important.

Although there has been a lot of variance bound in this area, few of them can guide experiment design directly, and we have actually made many attempts on finding appropriate variance bound. For example, the gradient of fourth-order term $\mathbb{E}[(t^T C t)^2]$ can be estimated directly through REINFORCE estimator [38] without need of upper bound. Nevertheless, the optimized covariance matrix tends to give a proportion of densely linked cluster pairs high correlation (near to 1), causing the covariance matrix to become nearly singular. This, in turn, leads to catastrophic gradient variance that can't be reduced by conventional technique such as [21]. In addition, weaker assumption on comparability between direct effect and interference is feasible. We can only demand there exists $\omega' > 0$ such that $\|h\|_2 \leq \omega' \gamma \|d\|_2$. This weaker assumption corresponds to spectral norm bound on the second order term $h^T \mathbb{E}[tt^T] h$, which is a looser bound and finally produces a empirically worse solution. This bound is also utilized in [17] and is expressed as the worst-case variance, which is concerned with maximizing a convex objective over bounded potential outcomes. The optimal value is taken when potential outcome is taken as the endpoint value of its domain, which is too conservative to have good empirical performance. The simulation result of IBR can support this claim, since similar worst-case variance is utilized in IBR.

Additionally, our work can be improved and extended in several aspects. Firstly, the optimization objective can be improved with a tighter but still optimizable upper bound on variance. Moreover, the situation of non-balanced treatment assignment, i.e., $\mathbb{E}[z_i] \neq \frac{1}{2}$, is worthy of exploration. Besides, the derivation for more complex estimator is kept explorable, e.g., the difference-in-means estimator, and HT estimator with exposure indicator. Our derivation is performed on the standard HT estimator, and structural assumption is introduced when the derivation encounters essential difficulties. In addition, the basic setting beyond neighborhood interference assumption also warrant further extension.

### Acknowledgments and Disclosure of Funding

The research was supported by the National Natural Science Foundation of China (No.72171131, 72133002); the Technology and Innovation Major Project of the Ministry of Science and Technology of China under Grants 2020AAA0108400 and 2020AAA0108403. We would like to thank Ruizhong Qiu for beneficial discussion, and anonymous reviewers for the helpful feedback.

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

# A  Proofs

## A.1  Proof of Proposition 1

Firstly, we notice that

$$\mathbb{E}[z_i] = \frac{1}{2} \quad \mathrm{Var}[z_i] = E[z_i] - E[z_i]^2 = \frac{1}{4} \tag{24}$$

Then we substitute $Y_i(\boldsymbol{z})$ in HT estimator

$$\hat{\tau} = \frac{1}{n} \sum_{i \in [n]} \left( \left( \frac{z_i}{\mathbb{E}[z_i]} - \frac{(1-z_i)}{\mathbb{E}[1-z_i]} \right) Y_i(z) \right) \tag{25}$$

with our potential outcome model and take expectation w.r.t. treatment assignments,

$$
\begin{aligned}
\mathbb{E}[\hat{\tau}] &= \frac{2}{n} \left( \sum_{i=1}^{n} \mathbb{E}[(z_i - (1 - z_i))Y_i(z)] \right) \\
&= \frac{2}{n} \sum_{i=1}^{n} \left( \alpha_i \mathbb{E}[2z_i - 1] + \beta_i \mathbb{E}[z_i^2] + \gamma \sum_{j \in N_i} \mathbb{E}[(2z_i - 1)z_j] \right) \\
&= \frac{2}{n} \sum_{i=1}^{n} \left( \beta_i \mathbb{E}[z_i] + 2\gamma \, \mathrm{Cov}[z_i, \sum_{j \in N_i} z_j] \right) \\
&= \frac{1}{n} \sum_{i=1}^{n} \left( \beta_i + 4\gamma \, \mathrm{Cov}[z_i, \sum_{j \in N_i} z_j] \right)
\end{aligned}
\tag{26}
$$

Moreover, we can calculate the GATE under our potential outcome model

$$\tau = \frac{1}{n} \sum_{i=1}^{n} (Y_i(\boldsymbol{1}) - Y_i(\boldsymbol{0})) = \frac{1}{n} \sum_{1=1}^{n} (\beta_i + \gamma d_i) \tag{27}$$

Hence, the bias of HT estimator is

$$
\begin{aligned}
E[\hat{\tau}] - \tau &= \frac{1}{n} \sum_{i=1}^{n} \left( 4\gamma \, \mathrm{Cov}[z_i, \sum_{j \in N_i} z_j] - \gamma d_i \right) \\
&= \frac{\gamma}{n} \sum_{i=1}^{n} \left( \frac{\mathrm{Cov}[z_i, \sum_{j \in N_i} z_j]}{\mathrm{Var}[z_i]} - d_i \right)
\end{aligned}
\tag{28}
$$

Then we derive the matrix form with cluster-level treatment vector $\boldsymbol{t}$.

$$
\begin{aligned}
\mathbb{E}[\hat{\tau}] - \tau &= \frac{\gamma}{n} \sum_{i=1}^{n} \left( \frac{\mathrm{Cov}[z_i, \sum_{k \in \mathcal{N}_i} z_k]}{\mathrm{Var}[z_i]} - d_i \right) \\
&= \frac{\gamma}{n} \left( 4 \sum_{i \neq j \in [K]} \mathrm{Cov}[t_i, t_j] \boldsymbol{C}_{ij} - \sum_{i \in [n]} d_i \right) \\
&= \frac{\gamma}{n} \left( 4 \sum_{i,j \in [K]} \mathrm{Cov}[t_i, t_j] \boldsymbol{C}_{ij} - \sum_{i,j \in [K]} \boldsymbol{C}_{ij} \right) \\
&= \frac{\gamma}{n} \left( 4 \, \mathrm{trace}(\boldsymbol{C} \, \mathrm{Cov}[\boldsymbol{t}]) - \sum_{i,j \in [K]} \boldsymbol{C}_{ij} \right)
\end{aligned}
\tag{29}
$$

## A.2 Proof of Proposition 2

Firstly we rewrite the estimator

$$\hat{\tau} = \frac{2}{n} \sum_{i \in [n]} \left( (\beta_i - \gamma d_i) z_i + 2\gamma \sum_{j \in N_i} z_i z_j \right)$$

$$= \frac{2}{n} \left( \sum_{i \in [n]} (\beta_i - \gamma d_i) z_i + 2\gamma \sum_{(j,k) \in \mathcal{E}} z_j z_k \right)$$

(30)

We then expand the variance directly, and get

$$\text{Var}[\hat{\tau}] = \frac{4}{n^2} \left( \sum_{i,j \in [n]} (\beta_i - \gamma d_i)(\beta_j - \gamma d_j) \text{Cov}[z_i, z_j] \right.$$

$$+ \sum_{i \in [n]} \sum_{(j,k) \in \mathcal{E}} 4\gamma(\beta_i - \gamma d_i) \text{Cov}[z_i, z_j z_k]$$

$$\left. + \sum_{(i,j) \in \mathcal{E}} \sum_{(k,l) \in \mathcal{E}} 4\gamma^2 \text{Cov}[z_i z_j, z_k z_l] \right)$$

(31)

then we write it as matrix form with cluster-level treatment vector $t$,

$$\hat{\tau} = \frac{2}{n}(\boldsymbol{h}^T t + 2\gamma t^T \boldsymbol{C} t)$$

(32)

and expand the variance as covariance, we get the desired format

$$\text{Var}[\hat{\tau}] = \frac{4}{n^2}(\text{trace}(\boldsymbol{h}\boldsymbol{h}^T \text{Cov}[t]) + 4\gamma \text{Cov}[\boldsymbol{h}^T t, t^T \boldsymbol{C} t]$$

$$+ 4\gamma^2 \text{Var}[t^T \boldsymbol{C} t])$$

(33)

## A.3 Proof of Proposition 3

Firstly we consider the estimator of matrix form again

$$\hat{\tau} = \frac{2}{n}(\boldsymbol{h}^T t + 2\gamma t^T \boldsymbol{C} t)$$

(34)

Since our assumption can't guarantee $\mathbb{E}[\hat{\tau}] > 0$ always hold, we drop the square of expectation term, namely

$$\text{Var}[\hat{\tau}] = \mathbb{E}[\hat{\tau}^2] - \mathbb{E}[\hat{\tau}]^2 \leq \mathbb{E}[\hat{\tau}^2]$$

(35)

Notice that all elements of matrix $E[tt^T]$ is non-negative, we have

$$\boldsymbol{h}^T \mathbb{E}[tt^T]\boldsymbol{h} \leq \omega^2 \boldsymbol{d}^T \mathbb{E}[tt^T]\boldsymbol{d} = \omega^2 \text{trace}(\boldsymbol{d}\boldsymbol{d}^T \mathbb{E}[tt^T])$$

(36)

Similarly, since all elements of matrix $\boldsymbol{C}$, namely, $\boldsymbol{C_{ij}}$ is also non-negative, we have

$$\mathbb{E}[(t^T \boldsymbol{C} t)^2] = \text{trace}(\mathbb{E}[\boldsymbol{C} tt^T \boldsymbol{C} tt^T])$$

$$\leq \text{trace}(\mathbb{E}[\boldsymbol{C}\mathbf{1}\mathbf{1}^T \boldsymbol{C} tt^T]$$

$$= \text{trace}(\boldsymbol{C}\mathbf{1}\mathbf{1}^T \boldsymbol{C}\mathbb{E}[tt^T])$$

(37)

In summary, we have

$$\text{Var}[\hat{\tau}] \leq \mathbb{E}[\hat{\tau}^2]$$

$$\leq \frac{8}{n^2}(\boldsymbol{h}^T \mathbb{E}[tt^T]\boldsymbol{h} + 4\gamma^2 \mathbb{E}[(t^T \boldsymbol{C} t)^2])$$

$$\leq \frac{8\gamma^2}{n^2} \left( \omega^2 \text{trace}(\boldsymbol{d}\boldsymbol{d}^T \mathbb{E}[tt^T]) + 4 \text{trace}(\boldsymbol{C}\mathbf{1}\mathbf{1}^T \boldsymbol{C}\mathbb{E}[tt^T]) \right)$$

(38)

By definition, we have

$$\boldsymbol{C}\mathbf{1} = \boldsymbol{d}$$

(39)

Thus the upper bound above is actually

$$\text{Var}[\hat{\tau}] \leq \frac{8\gamma^2(\omega^2+4)}{n^2}\left(\text{trace}(\boldsymbol{dd}^T\mathbb{E}[tt^T])\right) \tag{40}$$

At last, plug in following equation.

$$\mathbb{E}[tt^T] = \text{Cov}[t] + \frac{1}{4}\boldsymbol{11}^T \tag{41}$$

### A.4  Proof of Lemma 1

Here we provide an intuitive proof. Utilizing Box-Muller transformation or pure algebraic analysis are also feasible.

We consider $X$ and $Y$ is generated from multivariate Gaussian distribution,

$$X = \langle x, g \rangle \quad Y = \langle y, g \rangle \tag{42}$$

where $g \sim \mathcal{N}(0, I_n)$ and $x, y$ are two $n$-dim real vectors. Then we know that

$$\text{Cov}[X, Y] = \langle x, y \rangle \tag{43}$$

Moreover, we have

$$\mathbb{E}[\text{sgn}(X)] = 0 \quad \mathbb{E}[\text{sgn}(Y)] = 0 \tag{44}$$

thus

$$\text{Cov}[\text{sgn}(X), \text{sgn}(Y)] = \mathbb{E}[\text{sgn}(X)\,\text{sgn}(Y)] \tag{45}$$

Then we think geometrically that $\text{sgn}(\langle x, g \rangle)\,\text{sgn}(\langle y, g \rangle) > 0$ holds iff. $g$ lies above or below both of the hyperplanes that is orthogonal to $x$ and $y$ respectively.

Notice that the direction of $g$ is uniform, it follows that

$$\mathbb{P}(\text{sgn}(\langle x, g \rangle)\,\text{sgn}(\langle y, g \rangle) > 0) = \frac{2}{2\pi}(\pi - \arccos(\langle x, y \rangle)) \tag{46}$$

Now we put things together

$$\begin{aligned}
\mathbb{E}[\text{sgn}(X)\,\text{sgn}(Y)] &= \mathbb{P}(\text{sgn}(X)\,\text{sgn}(Y) > 0) - \mathbb{P}(\text{sgn}(X)\,\text{sgn}(Y) < 0) \\
&= 2\mathbb{P}(\text{sgn}(X)\,\text{sgn}(Y) > 0) - 1 \\
&= 2(\frac{1}{\pi}(\pi - \arccos(\langle x, y \rangle))) - 1 \\
&= 1 - \frac{2}{\pi}\arccos(\langle x, y \rangle) \\
&= \frac{2}{\pi}\arcsin(\langle x, y \rangle)
\end{aligned} \tag{47}$$

which gives the desired outcome.

## B  Simulation Details

### B.1  Methods

We consider following linear potential outcome model,

$$Y_i(\boldsymbol{z}) = \alpha + \beta \cdot z_i + c \cdot \frac{d_i}{\bar{d}} + \sigma \cdot \epsilon_i + \gamma\frac{\sum_{j \in N_i} z_j}{d_i} \tag{48}$$

and multiplicative model, which is a simplified version of that in [36], with removing a covariate.

$$Y_i(\boldsymbol{z}) = (\alpha + \sigma \cdot \epsilon_i) \cdot \frac{d_i}{\bar{d}} \cdot (1 + \beta z_i + \gamma\frac{\sum_{j \in N_i} z_j}{d_i}) \tag{49}$$

We choose to fix all parameters except for interference density, $\gamma$. Namely, we set $(\alpha, \beta, c, \sigma) = (1, 1, 0.5, 0.1)$ for both models, and set $\gamma \in \{0.5, 1, 2\}$ to construct three regimes. $\epsilon_i \sim \mathcal{N}(0, 1)$.

We set the clusters as given by Louvain algorithm with fixed random seed and resolution parameter as $2, 5, 10$.

We consider social network FB-Stanford3[32],[4] and FB-Cornell5[32].[5]. These two social networks provide the network topology, and we generate potential outcome for each unit with mentioned potential outcome models.

Besides our optimized covariance design (OCD), we implement following randomization schemes. We first consider two baselines, independent Bernoulli randomization (Ber) and complete randomization (CR), both of which are cluster-level. We also implement two adaptive schemes, rerandomized-adaptive randomization (ReAR) and pairwise-sequential randomization (PSR) [25, 31], which balance heuristic covariates adaptively and act as competitive baselines, since the average degree is considered as a covariate in these two methods and exactly appear in both two of our models, explicitly. Then we implement the independent block randomization (IBR) [5] and heuristic version IBR-p that creates blocks with size 2.

In the methods mentioned above, ReAR is the only one concerned with hyperparameters setting. We set $(q, B, \alpha) = (0.85, 400, 0.1)$, which corresponds to the recommendation in the original paper.

We estimate GATE with standard HT estimator and difference-in-means (DIM) estimator, where the latter refers to

$$\hat{\tau}_{DIM} = \sum_{i \in [n]} \left( \left( \frac{z_i}{\sum_{j \in [n]} z_j} - \frac{(1 - z_i)}{\sum_{j \in [n]} (1 - z_j)} \right) Y_i(z) \right) \tag{50}$$

To summarize, we provide the bias, standard deviation and MSE of two estimators under two potential outcome models, three $\gamma$ levels and two datasets. All of these three metrics are calculated by repetition of Monte Carlo simulation of randomization, 10,000 times. In the main paper we've presented the results of HT estimator on FB-Stanford3, and we'll present the detailed results in this section.

## B.2 Discussion on Estimators

Here we also provide discussion on another popular estimator that's considered in existing literature, which is the HT estimator with exposure indicator. We consider the exposure condition that's fully treated or fully controlled in 1-hop neighborhood here, and the corresponding indicator is defined as $\delta_i(z_0) = \mathbb{I}\{\sum_{j \in N_i} z_j = d_i z_0\}$. The estimator is

$$\hat{\tau}' = \frac{1}{n} \sum_{i \in [n]} \left( \left( \frac{\delta_i(1)}{\mathbb{E}[\delta_i(1)]} - \frac{\delta_i(0)}{\mathbb{E}[\delta_i(0)]} \right) Y_i(z) \right) \tag{51}$$

We don't consider this estimator not only because of it's high variance resulted from low effective sample size, and but also its high calculation cost. For calculating it in our repeated simulations, we need to estimate $\mathbb{E}[\delta_i(1)]$, the probability of fully treated, for each unit $i$. We denote the number of clusters a unit $i$ connects to by $c_i$. For a unit $i$ in the exterior of its cluster, namely, $c_i > 1$, such a quantity can very high in a social network, which is also decided by the resolution of clustering, we present an instance in figure 1.

Unfortunately, we must estimate such generalized propensity score [10] by Monte Carlo simulation for most of randomization schemes, even if it's just a little bit more complex than independent Bernoulli randomization, where such quantity can be calculated directly, $(1/2)^{c_i}$. For every simulation, we should visit every node and query the treatment assignment of its 1-hop neighborhood, whose time complexity is $O(|E|)$. Roughly, we need $2^{c_i}$ repetitions of randomization to achieve effective estimation on $\delta_i(1), \delta_i(0)$, which is too time-consuming to be acceptable for many units.

---

[4]Network topology data can be found in `https://networkrepository.com/socfb-Stanford3.php`
[5]`https://networkrepository.com/socfb-Cornell5.php`

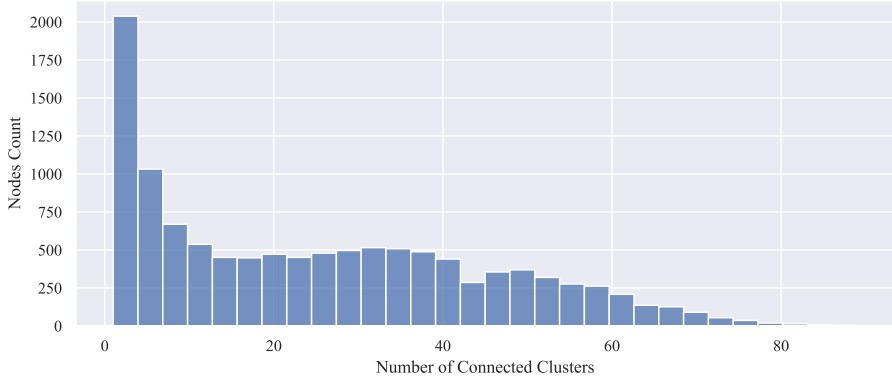

Figure 1: Distribution of $c_i$ with 95 clusters on FB-Stanford3

We stress the computational issue here since the resulted variance can be reduced by self-normalization, which corresponds to the Hájek estimator.

$$\hat{\tau}' = \sum_{i \in [n]} \left( \left( \frac{\frac{\delta_i(1)}{\mathbb{E}[\delta_i(1)]}}{\sum_{j \in [n]} \frac{\delta_j(1)}{\mathbb{E}[\delta_j(1)]}} \right) - \left( \frac{\frac{\delta_i(0)}{\mathbb{E}[\delta_i(0)]}}{\sum_{j \in [n]} \frac{\delta_j(0)}{\mathbb{E}[\delta_j(0)]}} \right) \right) Y_i(\boldsymbol{z}) \tag{52}$$

However, we argue that the computational cost can't be bypassed without further modification and restricts the practicality of such HT estimator with exposure indicator. [25] proposes the so-called cluster-adjusted estimator that assign the units on the exterior of same cluster the same modified propensity score, which reduces the computation complexity at the cost of unpredictable distortion of estimator, which can be viewed as heuristics.

Therefore, we choose to consider the standard HT estimator and DIM estimator in our simulation.

### B.3 Detailed Simulation Results

We present the results according to following sequence: HT-DIM estimator, linear-multiplicative model, 2-5-10 clustering resolution. In every table, we report the average bias, standard deviation (SD) and MSE of randomization schemes with three $\gamma$ levels. There are 24 tables in total, which are organized to demonstrate the advantage and robustness of our proposed method, optimized covariance design (OCD).

Table 3: Simulation results of **DIM** estimator under **linear** model with resolution **2** on FB-Stanford3

| gamma | | 0.5 | | | 1.0 | | | 2.0 | |
|---|---|---|---|---|---|---|---|---|---|
| metric | Bias | SD | MSE | Bias | SD | MSE | Bias | SD | MSE |
| method | | | | | | | | | |
| **Ber** | -0.247 | 0.121 | 0.076 | -0.497 | 0.122 | 0.262 | -0.988 | 0.131 | 0.995 |
| **CR** | -0.247 | 0.120 | 0.075 | -0.494 | 0.120 | 0.259 | -0.990 | 0.125 | 0.996 |
| **ReAR** | -0.248 | 0.063 | 0.066 | -0.482 | 0.065 | 0.237 | -0.958 | 0.062 | 0.923 |
| **PSR** | -0.243 | 0.059 | 0.063 | -0.486 | 0.059 | 0.240 | -0.974 | 0.062 | 0.954 |
| **IBR** | -0.248 | 0.107 | 0.073 | -0.493 | 0.109 | 0.255 | -0.985 | 0.112 | 0.985 |
| **IBR-p** | -0.244 | 0.090 | 0.068 | -0.490 | 0.092 | 0.249 | -0.980 | 0.094 | 0.969 |
| **OCD** | -0.195 | 0.067 | 0.043 | -0.388 | 0.068 | 0.156 | -0.776 | 0.072 | 0.608 |

Table 4: Simulation results of **HT** estimator under **linear** model with resolution **2** on FB-Stanford3

| gamma | | 0.5 | | | 1.0 | | | 2.0 | |
|---|---|---|---|---|---|---|---|---|---|
| metric | Bias | SD | MSE | Bias | SD | MSE | Bias | SD | MSE |
| method | | | | | | | | | |
| **Ber** | -0.228 | 1.028 | 1.110 | -0.489 | 1.148 | 1.558 | -0.966 | 1.370 | 2.810 |
| **CR** | -0.252 | 0.635 | 0.467 | -0.494 | 0.697 | 0.730 | -0.977 | 0.836 | 1.654 |
| **ReAR** | -0.236 | 0.190 | 0.092 | -0.486 | 0.173 | 0.267 | -0.940 | 0.173 | 0.914 |
| **PSR** | -0.241 | 0.233 | 0.113 | -0.481 | 0.253 | 0.296 | -0.972 | 0.297 | 1.033 |
| **IBR** | -0.247 | 0.226 | 0.112 | -0.494 | 0.252 | 0.308 | -0.987 | 0.302 | 1.067 |
| **IBR-p** | -0.246 | 0.154 | 0.085 | -0.489 | 0.168 | 0.268 | -0.981 | 0.203 | 1.006 |
| **OCD** | -0.189 | 0.219 | 0.084 | -0.385 | 0.252 | 0.212 | -0.775 | 0.312 | 0.699 |

Table 5: Simulation results of **DIM** estimator under **multiplicative** model with resolution **2** on FB-Stanford3

| gamma | | 0.5 | | | 1.0 | | | 2.0 | |
|---|---|---|---|---|---|---|---|---|---|
| metric | Bias | SD | MSE | Bias | SD | MSE | Bias | SD | MSE |
| method | | | | | | | | | |
| **Ber** | -0.294 | 0.419 | 0.263 | -0.603 | 0.479 | 0.593 | -1.209 | 0.605 | 1.830 |
| **CR** | -0.291 | 0.414 | 0.257 | -0.607 | 0.473 | 0.593 | -1.210 | 0.593 | 1.818 |
| **ReAR** | -0.324 | 0.209 | 0.149 | -0.573 | 0.237 | 0.386 | -1.274 | 0.317 | 1.725 |
| **PSR** | -0.307 | 0.208 | 0.138 | -0.619 | 0.236 | 0.440 | -1.236 | 0.297 | 1.618 |
| **IBR** | -0.305 | 0.374 | 0.234 | -0.605 | 0.427 | 0.549 | -1.220 | 0.536 | 1.778 |
| **IBR-p** | -0.304 | 0.315 | 0.192 | -0.611 | 0.366 | 0.508 | -1.214 | 0.457 | 1.684 |
| **OCD** | -0.250 | 0.232 | 0.117 | -0.507 | 0.265 | 0.328 | -1.006 | 0.333 | 1.123 |

Table 6: Simulation results of **HT** estimator under **multiplicative** model with resolution **2** on FB-Stanford3

| gamma | | 0.5 | | | 1.0 | | | 2.0 | |
|---|---|---|---|---|---|---|---|---|---|
| metric | Bias | SD | MSE | Bias | SD | MSE | Bias | SD | MSE |
| method | | | | | | | | | |
| **Ber** | -0.290 | 0.811 | 0.742 | -0.578 | 0.923 | 1.187 | -1.176 | 1.144 | 2.694 |
| **CR** | -0.301 | 0.519 | 0.361 | -0.605 | 0.582 | 0.705 | -1.213 | 0.739 | 2.019 |
| **ReAR** | -0.327 | 0.257 | 0.173 | -0.547 | 0.259 | 0.367 | -1.236 | 0.374 | 1.668 |
| **PSR** | -0.303 | 0.322 | 0.196 | -0.617 | 0.368 | 0.517 | -1.230 | 0.461 | 1.727 |
| **IBR** | -0.307 | 0.341 | 0.211 | -0.611 | 0.384 | 0.521 | -1.224 | 0.486 | 1.736 |
| **IBR-p** | -0.307 | 0.294 | 0.181 | -0.614 | 0.342 | 0.494 | -1.218 | 0.427 | 1.666 |
| **OCD** | -0.255 | 0.078 | 0.071 | -0.510 | 0.089 | 0.268 | -1.018 | 0.116 | 1.052 |

Table 7: Simulation results of **DIM** estimator under **linear** model with resolution **5** on FB-Stanford3

| gamma | | 0.5 | | | 1.0 | | | 2.0 | |
| metric
method | Bias | SD | MSE | Bias | SD | MSE | Bias | SD | MSE |
|---|---|---|---|---|---|---|---|---|---|
| **Ber** | -0.277 | 0.096 | 0.086 | -0.558 | 0.097 | 0.322 | -1.116 | 0.101 | 1.257 |
| **CR** | -0.279 | 0.095 | 0.087 | -0.556 | 0.095 | 0.319 | -1.117 | 0.097 | 1.259 |
| **ReAR** | -0.319 | 0.030 | 0.103 | -0.575 | 0.048 | 0.334 | -1.143 | 0.045 | 1.311 |
| **PSR** | -0.275 | 0.055 | 0.079 | -0.554 | 0.056 | 0.311 | -1.110 | 0.056 | 1.236 |
| **IBR** | -0.276 | 0.088 | 0.085 | -0.555 | 0.089 | 0.316 | -1.110 | 0.090 | 1.241 |
| **IBR-p** | -0.209 | 0.074 | 0.049 | -0.490 | 0.074 | 0.246 | -1.049 | 0.074 | 1.106 |
| **OCD** | -0.194 | 0.102 | 0.048 | -0.390 | 0.102 | 0.163 | -0.782 | 0.104 | 0.623 |

Table 8: Simulation results of **HT** estimator under **linear** model with resolution **5** on FB-Stanford3

| gamma | | 0.5 | | | 1.0 | | | 2.0 | |
| metric
method | Bias | SD | MSE | Bias | SD | MSE | Bias | SD | MSE |
|---|---|---|---|---|---|---|---|---|---|
| **Ber** | -0.276 | 0.666 | 0.521 | -0.535 | 0.749 | 0.847 | -1.078 | 0.910 | 1.991 |
| **CR** | -0.316 | 0.488 | 0.338 | -0.610 | 0.540 | 0.664 | -1.159 | 0.664 | 1.786 |
| **ReAR** | -0.063 | 0.084 | 0.011 | -0.410 | 0.219 | 0.216 | -0.873 | 0.215 | 0.809 |
| **PSR** | -0.272 | 0.222 | 0.123 | -0.544 | 0.250 | 0.359 | -1.108 | 0.310 | 1.326 |
| **IBR** | -0.272 | 0.276 | 0.150 | -0.553 | 0.313 | 0.404 | -1.103 | 0.390 | 1.370 |
| **IBR-p** | -0.694 | 0.064 | 0.486 | -1.025 | 0.074 | 1.057 | -1.688 | 0.097 | 2.861 |
| **OCD** | -0.198 | 0.386 | 0.189 | -0.391 | 0.439 | 0.346 | -0.772 | 0.547 | 0.896 |

Table 9: Simulation results of **DIM** estimator under **multiplicative** model with resolution **5** on FB-Stanford3

| gamma | | 0.5 | | | 1.0 | | | 2.0 | |
| metric
method | Bias | SD | MSE | Bias | SD | MSE | Bias | SD | MSE |
|---|---|---|---|---|---|---|---|---|---|
| **Ber** | -0.338 | 0.333 | 0.226 | -0.687 | 0.387 | 0.622 | -1.382 | 0.480 | 2.142 |
| **CR** | -0.340 | 0.333 | 0.226 | -0.680 | 0.387 | 0.613 | -1.382 | 0.477 | 2.138 |
| **ReAR** | -0.417 | 0.171 | 0.204 | -0.860 | 0.163 | 0.767 | -1.485 | 0.228 | 2.260 |
| **PSR** | -0.336 | 0.192 | 0.150 | -0.680 | 0.219 | 0.511 | -1.375 | 0.277 | 1.968 |
| **IBR** | -0.344 | 0.310 | 0.214 | -0.680 | 0.351 | 0.586 | -1.379 | 0.439 | 2.097 |
| **IBR-p** | -0.098 | 0.272 | 0.084 | -0.407 | 0.312 | 0.263 | -1.025 | 0.392 | 1.205 |
| **OCD** | -0.241 | 0.357 | 0.185 | -0.497 | 0.407 | 0.413 | -0.994 | 0.511 | 1.249 |

Table 10: Simulation results of **HT** estimator under **multiplicative** model with resolution **5** on FB-Stanford3

| gamma | | 0.5 | | | 1.0 | | | 2.0 | |
| metric
method | Bias | SD | MSE | Bias | SD | MSE | Bias | SD | MSE |
|---|---|---|---|---|---|---|---|---|---|
| **Ber** | -0.340 | 0.463 | 0.330 | -0.683 | 0.523 | 0.740 | -1.366 | 0.656 | 2.297 |
| **CR** | -0.383 | 0.289 | 0.230 | -0.733 | 0.328 | 0.645 | -1.430 | 0.412 | 2.215 |
| **ReAR** | -0.289 | 0.138 | 0.102 | -0.685 | 0.121 | 0.484 | -1.288 | 0.230 | 1.712 |
| **PSR** | -0.335 | 0.128 | 0.129 | -0.684 | 0.146 | 0.490 | -1.373 | 0.182 | 1.919 |
| **IBR** | -0.346 | 0.180 | 0.153 | -0.694 | 0.207 | 0.525 | -1.387 | 0.259 | 1.992 |
| **IBR-p** | -0.488 | 0.190 | 0.275 | -0.852 | 0.217 | 0.774 | -1.581 | 0.271 | 2.574 |
| **OCD** | -0.255 | 0.049 | 0.068 | -0.508 | 0.057 | 0.262 | -1.020 | 0.069 | 1.045 |

Table 11: Simulation results of **DIM** estimator under **linear** model with resolution **10** on FB-Stanford3

| gamma | 0.5 | | | 1.0 | | | 2.0 | | |
|--------|------|------|------|------|------|------|------|------|------|
| metric | Bias | SD | MSE | Bias | SD | MSE | Bias | SD | MSE |
| method | | | | | | | | | |
| **Ber** | -0.298 | 0.076 | 0.095 | -0.595 | 0.077 | 0.361 | -1.191 | 0.080 | 1.426 |
| **CR** | -0.297 | 0.076 | 0.095 | -0.595 | 0.076 | 0.360 | -1.191 | 0.078 | 1.427 |
| **ReAR** | -0.296 | 0.022 | 0.088 | -0.589 | 0.019 | 0.348 | -1.191 | 0.029 | 1.422 |
| **PSR** | -0.297 | 0.035 | 0.090 | -0.596 | 0.036 | 0.357 | -1.193 | 0.039 | 1.425 |
| **IBR** | -0.296 | 0.054 | 0.091 | -0.593 | 0.055 | 0.356 | -1.188 | 0.058 | 1.415 |
| **IBR-p** | -0.297 | 0.049 | 0.091 | -0.593 | 0.049 | 0.354 | -1.186 | 0.052 | 1.411 |
| **OCD** | -0.190 | 0.115 | 0.050 | -0.384 | 0.116 | 0.161 | -0.772 | 0.118 | 0.611 |

Table 12: Simulation results of **HT** estimator under **linear** model with resolution **10** on FB-Stanford3

| gamma | 0.5 | | | 1.0 | | | 2.0 | | |
|--------|------|------|------|------|------|------|------|------|------|
| metric | Bias | SD | MSE | Bias | SD | MSE | Bias | SD | MSE |
| method | | | | | | | | | |
| **Ber** | -0.293 | 0.521 | 0.358 | -0.588 | 0.584 | 0.688 | -1.178 | 0.709 | 1.893 |
| **CR** | -0.292 | 0.409 | 0.253 | -0.586 | 0.459 | 0.554 | -1.177 | 0.562 | 1.702 |
| **ReAR** | -0.393 | 0.227 | 0.206 | -0.700 | 0.251 | 0.554 | -1.317 | 0.303 | 1.829 |
| **PSR** | -0.295 | 0.235 | 0.143 | -0.587 | 0.264 | 0.415 | -1.179 | 0.323 | 1.496 |
| **IBR** | -0.298 | 0.273 | 0.164 | -0.593 | 0.308 | 0.447 | -1.181 | 0.380 | 1.541 |
| **IBR-p** | -0.294 | 0.232 | 0.141 | -0.596 | 0.261 | 0.423 | -1.185 | 0.318 | 1.507 |
| **OCD** | -0.198 | 0.411 | 0.209 | -0.388 | 0.469 | 0.371 | -0.764 | 0.585 | 0.926 |

Table 13: Simulation results of **DIM** estimator under **multiplicative** model with resolution **10** on FB-Stanford3

| gamma | 0.5 | | | 1.0 | | | 2.0 | | |
|--------|------|------|------|------|------|------|------|------|------|
| metric | Bias | SD | MSE | Bias | SD | MSE | Bias | SD | MSE |
| method | | | | | | | | | |
| **Ber** | -0.369 | 0.264 | 0.206 | -0.731 | 0.301 | 0.626 | -1.475 | 0.375 | 2.318 |
| **CR** | -0.363 | 0.265 | 0.202 | -0.737 | 0.303 | 0.636 | -1.470 | 0.380 | 2.307 |
| **ReAR** | -0.351 | 0.061 | 0.127 | -0.715 | 0.095 | 0.520 | -1.480 | 0.121 | 2.207 |
| **PSR** | -0.369 | 0.122 | 0.151 | -0.738 | 0.141 | 0.565 | -1.481 | 0.176 | 2.227 |
| **IBR** | -0.367 | 0.190 | 0.171 | -0.737 | 0.218 | 0.592 | -1.476 | 0.271 | 2.254 |
| **IBR-p** | -0.364 | 0.168 | 0.161 | -0.742 | 0.192 | 0.588 | -1.480 | 0.244 | 2.253 |
| **OCD** | -0.247 | 0.402 | 0.223 | -0.501 | 0.459 | 0.463 | -1.013 | 0.577 | 1.360 |

Table 14: Simulation results of **HT** estimator under **multiplicative** model with resolution **10** on FB-Stanford3

| gamma | 0.5 | | | 1.0 | | | 2.0 | | |
|--------|------|------|------|------|------|------|------|------|------|
| metric | Bias | SD | MSE | Bias | SD | MSE | Bias | SD | MSE |
| method | | | | | | | | | |
| **Ber** | -0.365 | 0.348 | 0.255 | -0.736 | 0.394 | 0.698 | -1.475 | 0.493 | 2.421 |
| **CR** | -0.368 | 0.235 | 0.191 | -0.744 | 0.274 | 0.629 | -1.477 | 0.336 | 2.297 |
| **ReAR** | -0.402 | 0.178 | 0.194 | -0.809 | 0.174 | 0.685 | -1.548 | 0.226 | 2.450 |
| **PSR** | -0.366 | 0.134 | 0.152 | -0.738 | 0.153 | 0.569 | -1.479 | 0.192 | 2.227 |
| **IBR** | -0.369 | 0.155 | 0.161 | -0.737 | 0.178 | 0.576 | -1.484 | 0.221 | 2.252 |
| **IBR-p** | -0.368 | 0.163 | 0.163 | -0.739 | 0.185 | 0.581 | -1.482 | 0.232 | 2.252 |
| **OCD** | -0.258 | 0.040 | 0.069 | -0.517 | 0.050 | 0.271 | -1.034 | 0.054 | 1.073 |

Table 15: Simulation results of **DIM** estimator under **linear** model with resolution **2** on FB-Cornell5

| gamma | | 0.5 | | | 1.0 | | | 2.0 | |
|---|---|---|---|---|---|---|---|---|---|
| metric | Bias | SD | MSE | Bias | SD | MSE | Bias | SD | MSE |
| **method** | | | | | | | | | |
| **Ber** | -0.240 | 0.092 | 0.066 | -0.480 | 0.093 | 0.239 | -0.960 | 0.103 | 0.933 |
| **CR** | -0.238 | 0.090 | 0.065 | -0.480 | 0.092 | 0.239 | -0.958 | 0.100 | 0.929 |
| **ReAR** | -0.239 | 0.019 | 0.058 | -0.469 | 0.026 | 0.221 | -0.925 | 0.032 | 0.858 |
| **PSR** | -0.235 | 0.056 | 0.059 | -0.470 | 0.056 | 0.225 | -0.944 | 0.058 | 0.894 |
| **IBR** | -0.235 | 0.091 | 0.064 | -0.477 | 0.093 | 0.237 | -0.950 | 0.096 | 0.913 |
| **IBR-p** | -0.235 | 0.103 | 0.066 | -0.471 | 0.104 | 0.234 | -0.944 | 0.108 | 0.903 |
| **OCD** | -0.188 | 0.059 | 0.039 | -0.374 | 0.060 | 0.144 | -0.751 | 0.070 | 0.569 |

Table 16: Simulation results of **HT** estimator under **linear** model with resolution **2** on FB-Cornell5

| gamma | | 0.5 | | | 1.0 | | | 2.0 | |
|---|---|---|---|---|---|---|---|---|---|
| metric | Bias | SD | MSE | Bias | SD | MSE | Bias | SD | MSE |
| **method** | | | | | | | | | |
| **Ber** | -0.228 | 1.087 | 1.234 | -0.456 | 1.240 | 1.746 | -0.913 | 1.469 | 2.993 |
| **CR** | -0.228 | 0.913 | 0.885 | -0.440 | 1.018 | 1.230 | -0.914 | 1.231 | 2.350 |
| **ReAR** | -0.209 | 0.110 | 0.056 | -0.469 | 0.128 | 0.237 | -0.965 | 0.135 | 0.950 |
| **PSR** | -0.235 | 0.146 | 0.077 | -0.473 | 0.164 | 0.251 | -0.940 | 0.202 | 0.926 |
| **IBR** | -0.232 | 0.530 | 0.335 | -0.465 | 0.595 | 0.571 | -0.942 | 0.720 | 1.407 |
| **IBR-p** | -0.233 | 0.180 | 0.087 | -0.472 | 0.209 | 0.268 | -0.939 | 0.263 | 0.951 |
| **OCD** | -0.184 | 0.213 | 0.079 | -0.376 | 0.241 | 0.200 | -0.752 | 0.302 | 0.657 |

Table 17: Simulation results of **DIM** estimator under **multiplicative** model with resolution **2** on FB-Cornell5

| gamma | | 0.5 | | | 1.0 | | | 2.0 | |
|---|---|---|---|---|---|---|---|---|---|
| metric | Bias | SD | MSE | Bias | SD | MSE | Bias | SD | MSE |
| **method** | | | | | | | | | |
| **Ber** | -0.260 | 0.318 | 0.169 | -0.546 | 0.363 | 0.430 | -1.075 | 0.461 | 1.369 |
| **CR** | -0.261 | 0.317 | 0.169 | -0.533 | 0.362 | 0.416 | -1.080 | 0.457 | 1.375 |
| **ReAR** | -0.252 | 0.064 | 0.068 | -0.515 | 0.090 | 0.273 | -1.073 | 0.135 | 1.172 |
| **PSR** | -0.268 | 0.190 | 0.108 | -0.536 | 0.223 | 0.338 | -1.069 | 0.276 | 1.221 |
| **IBR** | -0.268 | 0.319 | 0.174 | -0.535 | 0.368 | 0.422 | -1.074 | 0.453 | 1.359 |
| **IBR-p** | -0.258 | 0.362 | 0.198 | -0.526 | 0.416 | 0.450 | -1.050 | 0.514 | 1.367 |
| **OCD** | -0.215 | 0.201 | 0.087 | -0.428 | 0.228 | 0.235 | -0.866 | 0.291 | 0.835 |

Table 18: Simulation results of **HT** estimator under **multiplicative** model with resolution **2** on FB-Cornell5

| gamma | | 0.5 | | | 1.0 | | | 2.0 | |
|---|---|---|---|---|---|---|---|---|---|
| metric | Bias | SD | MSE | Bias | SD | MSE | Bias | SD | MSE |
| **method** | | | | | | | | | |
| **Ber** | -0.260 | 0.853 | 0.795 | -0.517 | 0.973 | 1.214 | -1.020 | 1.213 | 2.514 |
| **CR** | -0.265 | 0.700 | 0.561 | -0.536 | 0.801 | 0.929 | -1.053 | 1.011 | 2.133 |
| **ReAR** | -0.267 | 0.057 | 0.075 | -0.515 | 0.076 | 0.271 | -1.073 | 0.106 | 1.164 |
| **PSR** | -0.269 | 0.151 | 0.095 | -0.539 | 0.176 | 0.322 | -1.070 | 0.218 | 1.194 |
| **IBR** | -0.264 | 0.419 | 0.246 | -0.526 | 0.487 | 0.515 | -1.074 | 0.614 | 1.531 |
| **IBR-p** | -0.264 | 0.243 | 0.129 | -0.534 | 0.277 | 0.362 | -1.066 | 0.344 | 1.255 |
| **OCD** | -0.218 | 0.059 | 0.051 | -0.436 | 0.068 | 0.195 | -0.873 | 0.085 | 0.770 |

Table 19: Simulation results of **DIM** estimator under **linear** model with resolution **5** on FB-Cornell5

| gamma | 0.5 | | | 1.0 | | | 2.0 | | |
|---|---|---|---|---|---|---|---|---|---|
| metric
method | Bias | SD | MSE | Bias | SD | MSE | Bias | SD | MSE |
| **Ber** | -0.281 | 0.070 | 0.084 | -0.560 | 0.070 | 0.319 | -1.124 | 0.074 | 1.269 |
| **CR** | -0.280 | 0.069 | 0.084 | -0.561 | 0.069 | 0.320 | -1.123 | 0.072 | 1.268 |
| **ReAR** | -0.282 | 0.048 | 0.082 | -0.559 | 0.045 | 0.315 | -1.113 | 0.035 | 1.241 |
| **PSR** | -0.274 | 0.052 | 0.078 | -0.552 | 0.053 | 0.308 | -1.106 | 0.055 | 1.227 |
| **IBR** | -0.280 | 0.069 | 0.083 | -0.560 | 0.070 | 0.319 | -1.120 | 0.073 | 1.261 |
| **IBR-p** | -0.278 | 0.086 | 0.085 | -0.558 | 0.086 | 0.319 | -1.113 | 0.088 | 1.247 |
| **OCD** | -0.196 | 0.120 | 0.053 | -0.393 | 0.120 | 0.169 | -0.784 | 0.125 | 0.631 |

Table 20: Simulation results of **HT** estimator under **linear** model with resolution **5** on FB-Cornell5

| gamma | 0.5 | | | 1.0 | | | 2.0 | | |
|---|---|---|---|---|---|---|---|---|---|
| metric
method | Bias | SD | MSE | Bias | SD | MSE | Bias | SD | MSE |
| **Ber** | -0.268 | 0.676 | 0.529 | -0.551 | 0.755 | 0.874 | -1.106 | 0.918 | 2.067 |
| **CR** | -0.279 | 0.558 | 0.389 | -0.553 | 0.622 | 0.693 | -1.092 | 0.748 | 1.753 |
| **ReAR** | -0.305 | 0.376 | 0.234 | -0.608 | 0.427 | 0.553 | -1.243 | 0.516 | 1.812 |
| **PSR** | -0.275 | 0.373 | 0.215 | -0.547 | 0.419 | 0.475 | -1.088 | 0.511 | 1.446 |
| **IBR** | -0.275 | 0.409 | 0.243 | -0.556 | 0.461 | 0.522 | -1.100 | 0.557 | 1.521 |
| **IBR-p** | -0.274 | 0.332 | 0.185 | -0.550 | 0.377 | 0.445 | -1.112 | 0.466 | 1.454 |
| **OCD** | -0.194 | 0.433 | 0.225 | -0.385 | 0.490 | 0.389 | -0.781 | 0.616 | 0.991 |

Table 21: Simulation results of **DIM** estimator under **multiplicative** model with resolution **5** on FB-Cornell5

| gamma | 0.5 | | | 1.0 | | | 2.0 | | |
|---|---|---|---|---|---|---|---|---|---|
| metric
method | Bias | SD | MSE | Bias | SD | MSE | Bias | SD | MSE |
| **Ber** | -0.314 | 0.239 | 0.156 | -0.634 | 0.276 | 0.479 | -1.269 | 0.344 | 1.729 |
| **CR** | -0.313 | 0.238 | 0.154 | -0.630 | 0.273 | 0.472 | -1.273 | 0.344 | 1.741 |
| **ReAR** | -0.330 | 0.157 | 0.134 | -0.623 | 0.164 | 0.416 | -1.256 | 0.181 | 1.613 |
| **PSR** | -0.309 | 0.180 | 0.128 | -0.622 | 0.207 | 0.430 | -1.252 | 0.262 | 1.636 |
| **IBR** | -0.312 | 0.244 | 0.157 | -0.635 | 0.277 | 0.481 | -1.266 | 0.349 | 1.725 |
| **IBR-p** | -0.305 | 0.298 | 0.182 | -0.625 | 0.342 | 0.508 | -1.251 | 0.427 | 1.747 |
| **OCD** | -0.212 | 0.417 | 0.219 | -0.428 | 0.477 | 0.412 | -0.868 | 0.600 | 1.114 |

Table 22: Simulation results of **HT** estimator under **multiplicative** model with resolution **5** on FB-Cornell5

| gamma | 0.5 | | | 1.0 | | | 2.0 | | |
|---|---|---|---|---|---|---|---|---|---|
| metric
method | Bias | SD | MSE | Bias | SD | MSE | Bias | SD | MSE |
| **Ber** | -0.302 | 0.496 | 0.338 | -0.629 | 0.548 | 0.696 | -1.245 | 0.690 | 2.027 |
| **CR** | -0.320 | 0.375 | 0.243 | -0.632 | 0.430 | 0.586 | -1.262 | 0.539 | 1.885 |
| **ReAR** | -0.310 | 0.230 | 0.150 | -0.623 | 0.242 | 0.447 | -1.308 | 0.360 | 1.841 |
| **PSR** | -0.307 | 0.203 | 0.136 | -0.618 | 0.233 | 0.436 | -1.252 | 0.290 | 1.654 |
| **IBR** | -0.318 | 0.239 | 0.159 | -0.632 | 0.277 | 0.477 | -1.274 | 0.349 | 1.747 |
| **IBR-p** | -0.315 | 0.148 | 0.121 | -0.634 | 0.168 | 0.431 | -1.265 | 0.211 | 1.647 |
| **OCD** | -0.225 | 0.042 | 0.053 | -0.451 | 0.050 | 0.206 | -0.904 | 0.059 | 0.821 |

Table 23: Simulation results of **DIM** estimator under **linear** model with resolution **10** on FB-Cornell5

| gamma | | 0.5 | | | 1.0 | | | 2.0 | |
|---|---|---|---|---|---|---|---|---|---|
| metric | Bias | SD | MSE | Bias | SD | MSE | Bias | SD | MSE |
| method | | | | | | | | | |
| **Ber** | -0.312 | 0.038 | 0.099 | -0.624 | 0.040 | 0.392 | -1.249 | 0.044 | 1.563 |
| **CR** | -0.312 | 0.039 | 0.099 | -0.624 | 0.040 | 0.392 | -1.249 | 0.043 | 1.563 |
| **ReAR** | -0.308 | 0.031 | 0.096 | -0.625 | 0.024 | 0.392 | -1.253 | 0.029 | 1.572 |
| **PSR** | -0.314 | 0.035 | 0.100 | -0.625 | 0.035 | 0.393 | -1.252 | 0.038 | 1.569 |
| **IBR** | -0.312 | 0.036 | 0.099 | -0.625 | 0.038 | 0.392 | -1.250 | 0.041 | 1.566 |
| **IBR-p** | -0.311 | 0.036 | 0.098 | -0.623 | 0.036 | 0.390 | -1.246 | 0.042 | 1.555 |
| **OCD** | -0.188 | 0.098 | 0.045 | -0.374 | 0.100 | 0.150 | -0.749 | 0.106 | 0.574 |

Table 24: Simulation results of **HT** estimator under **linear** model with resolution **10** on FB-Cornell5

| gamma | | 0.5 | | | 1.0 | | | 2.0 | |
|---|---|---|---|---|---|---|---|---|---|
| metric | Bias | SD | MSE | Bias | SD | MSE | Bias | SD | MSE |
| method | | | | | | | | | |
| **Ber** | -0.311 | 0.388 | 0.247 | -0.621 | 0.437 | 0.576 | -1.237 | 0.517 | 1.799 |
| **CR** | -0.311 | 0.264 | 0.167 | -0.623 | 0.290 | 0.473 | -1.243 | 0.345 | 1.665 |
| **ReAR** | -0.306 | 0.069 | 0.099 | -0.618 | 0.087 | 0.390 | -1.238 | 0.098 | 1.544 |
| **PSR** | -0.310 | 0.067 | 0.101 | -0.622 | 0.076 | 0.394 | -1.245 | 0.094 | 1.560 |
| **IBR** | -0.312 | 0.105 | 0.108 | -0.627 | 0.116 | 0.407 | -1.250 | 0.139 | 1.583 |
| **IBR-p** | -0.311 | 0.065 | 0.102 | -0.623 | 0.073 | 0.395 | -1.246 | 0.090 | 1.562 |
| **OCD** | -0.182 | 0.345 | 0.152 | -0.375 | 0.397 | 0.299 | -0.749 | 0.495 | 0.807 |

Table 25: Simulation results of **DIM** estimator under **multiplicative** model with resolution **10** on FB-Cornell5

| gamma | | 0.5 | | | 1.0 | | | 2.0 | |
|---|---|---|---|---|---|---|---|---|---|
| metric | Bias | SD | MSE | Bias | SD | MSE | Bias | SD | MSE |
| method | | | | | | | | | |
| **Ber** | -0.354 | 0.136 | 0.144 | -0.702 | 0.155 | 0.518 | -1.404 | 0.195 | 2.012 |
| **CR** | -0.352 | 0.138 | 0.143 | -0.702 | 0.154 | 0.517 | -1.407 | 0.194 | 2.019 |
| **ReAR** | -0.354 | 0.077 | 0.131 | -0.699 | 0.099 | 0.499 | -1.394 | 0.142 | 1.964 |
| **PSR** | -0.355 | 0.119 | 0.141 | -0.708 | 0.136 | 0.520 | -1.411 | 0.174 | 2.023 |
| **IBR** | -0.352 | 0.126 | 0.141 | -0.704 | 0.145 | 0.517 | -1.411 | 0.182 | 2.025 |
| **IBR-p** | -0.350 | 0.124 | 0.138 | -0.703 | 0.142 | 0.516 | -1.403 | 0.178 | 2.003 |
| **OCD** | -0.215 | 0.343 | 0.164 | -0.438 | 0.395 | 0.348 | -0.878 | 0.490 | 1.013 |

Table 26: Simulation results of **HT** estimator under **multiplicative** model with resolution **10** on FB-Cornell5

| gamma | | 0.5 | | | 1.0 | | | 2.0 | |
|---|---|---|---|---|---|---|---|---|---|
| metric | Bias | SD | MSE | Bias | SD | MSE | Bias | SD | MSE |
| method | | | | | | | | | |
| **Ber** | -0.355 | 0.325 | 0.232 | -0.698 | 0.370 | 0.625 | -1.396 | 0.469 | 2.171 |
| **CR** | -0.349 | 0.238 | 0.179 | -0.704 | 0.266 | 0.567 | -1.401 | 0.335 | 2.075 |
| **ReAR** | -0.353 | 0.077 | 0.131 | -0.708 | 0.090 | 0.511 | -1.399 | 0.131 | 1.976 |
| **PSR** | -0.353 | 0.101 | 0.135 | -0.707 | 0.116 | 0.513 | -1.408 | 0.147 | 2.005 |
| **IBR** | -0.351 | 0.141 | 0.143 | -0.703 | 0.162 | 0.521 | -1.409 | 0.207 | 2.029 |
| **IBR-p** | -0.349 | 0.107 | 0.134 | -0.703 | 0.122 | 0.509 | -1.404 | 0.154 | 1.997 |
| **OCD** | -0.224 | 0.038 | 0.052 | -0.448 | 0.041 | 0.203 | -0.895 | 0.053 | 0.805 |

