# OpenReview forum: "Optimized Covariance Design for AB Test on Social Network under Interference"
_NeurIPS.cc/2023/Conference — NeurIPS 2023 poster_

### Official Review · Reviewer_EFcj · 2023-07-03

**Soundness:** 2 fair
**Presentation:** 3 good
**Contribution:** 2 fair
**Rating:** 6
**Confidence:** 3

**Summary:**

This paper focused on designing a randomization scheme at the cluster level for the A/B test. It proposed and derived an upper bound for the MSE of the HT estimator, which was targeted to be minimized to optimize the experiment design. This article treated the covariance matrix of the treatment vector as the decision variable in optimization to represent a class of design. Besides, it proposed to adopt projected gradient descent to guarantee the validity of the optimized covariance matrix in order to generate a randomization scheme that supports legitimate sampling. Also, it stressed the significance of boas and attempted to balance bias and variance better. Systematic simulations on a real-time social network were conducted. Experiment outcomes were analyzed and different methods were compared to show the effectiveness of the newly proposed method in this article.

**Strengths:**

1. This paper is generally easy to follow.
2. The problem is well formulated in a new perspective, with detailed analysis and deduction.
3. The effectiveness of the proposed method is verified in the experiment. Horizontal comparisons with existing methods are sufficient and effective.

**Weaknesses:**

1. Insufficient background introduction. Different concepts in the area should be introduced more specifically and it is better to point out their connections.
2. Unclear notation definitions and explanations, leading to confusion in understanding.
3. Experiments were not comprehensive, or at least not comprehensively expressed.

**Questions:**

1. Where is the definition of di that appeared below line 150 in assumption 1? (Does it refer to the degree of node i?) It is hard to read when definitions are missed or distributed in different parts. It is better to summarize all of them into one table with clear definitions.
2. In section 2.1, line 132, what are the certain community detection algorithms? It is better to list out some works with references.
3. The experiment part seemed to be written in a hurry. Even though necessary comparison methods were involved, many other essential details such as the basic information about the dataset (i.e., edges, nodes, degrees, etc.) were missing. Also, it seems that experiments were only implemented on one dataset, which made the experiment outcome not convincing enough.
4. What is the real-world application of this method? Please offer some direct applications such as the areas that may benefit from your method. This is important since your derivation used different assumptions when dealing with difficult parts in the optimization. Will the assumptions affect its performance in general? Is the proposed method only workable under a specific scene and what is the case?

**Limitations:**

Limitations of this article were clearly expressed in this article. For instance, the upper bound should be tighter in the optimization. And different other cases like the non-balanced treatment assignment have not been considered yet. The basic setting beyond nationhood interference assumption warrants a further extension. Besides, assumptions may hinder the method from applying to a more generalized case.

---

> ### Author Rebuttal · Authors · 2023-08-05
>
> First of all, thanks for your time and efforts in the reviewing process, and we appreciate your approval on the problem formulation and content presentation of our work. We address your concerns as follows.
>
> ## Not comprehensive experiments
>
> We conduct an array of simulations with a substantial portion of the detailed results presented in the appendix. We specifically focus on highlighting the advantages of our proposed OCD method, particularly in scenarios where interference intensity, represented by γ, is elevated. We think our simulation study is systematic and well-formulated to demonstrate the efficacy and robustness of our proposed method.
>
> ## Responses to questions
>
> 1. **Clarification on notation**: Thank you for highlighting the oversight regarding the notation $d_i$.Yes, $d_i$ indicates the degree of unit $i$. We acknowledge that this definition is introduced in the linear potential outcome model originally within the introduction, which was finally deleted due to limited space. We will certainly address this in the upcoming version. Rest assured, other notations in the paper are accompanied by clear definitions or appropriate citations, ensuring overall clarity.
>
> 2. **Community detection algorithm**: We appreciate your attention to detail, and we'll add additional notes in line 132 for better clarity. Actually, we've mentioned and cited the classical community detection algorithm (Louvain) in the introduction (line 44), and this algorithm is also mentioned in our simulation study.
>
> 3. **Network statistics and dataset results**: We think network statistics is relatively unnecessary for the body of this paper, and we provide the accessible links to the panel of such network topology in the footnote of page 7 (and page 4 in the appendix), where most of statistics of network topology are provided. Secondly, we provide the results of two datasets in total, in the appendix, which is also mentioned in line 303.
>
> 4. **Simulation for method applicability and scalability**: We think the comprehensive simulation has provided adequate evidence for the applicability of our method on different scenes, which includes different interference intensities, different clustering resolutions (corresponds to cluster numbers), different estimators, different potential outcome models, and different datasets. As mentioned in the introduction, experiment design for AB test on social network is widely used for social platforms such as Facebook, LinkedIn, and WeChat. The deployment of every new trait for social platform must come through a series of AB tests with increasing scale, which is a classical and important scenario for our method and mentioned in the introduction. Moreover, we've argued that the gradient-based algorithm for our continuous optimization problem is scalable for practical scenes and confirmed by our industrial collaborator.

---

> > ### Comment · Reviewer_EFcj · 2023-08-18
> > **Thanks for your reply.**
> >
> > I have carefully read your response. The supplementary experiments complete the work and it solves of my previous doubts. I believe this work will contribute to the community and would like to raise my rating to 6.

---

> > > ### Author Response · Authors · 2023-08-19
> > >
> > > Thank you again for your approval on this paper. We'll present the simulation result in a clearer way in the upcoming version.

---

### Official Review · Reviewer_1xr3 · 2023-07-04

**Soundness:** 3 good
**Presentation:** 3 good
**Contribution:** 3 good
**Rating:** 6
**Confidence:** 5

**Summary:**

In this paper, authors present a novel experimental design to be used for randomized experiments under network interference.
Authors begin by considering a baseline adjusted Horvitz--Thompson estimator (which crucially requires knowledge of $Y_i(\mathbf{0})$) and a pre-specified clusterings.
Given these two things, authors derive a bound on the mean squared error of the estimator under an arbitrary cluster design.
Motivated by a Grothendeick identity, authors propose an experimental design where a normal vector is sampled from a fixed covariance matrix $\Sigma$ and the signs of the vector become treatment assignment.
Authors propose to select an experimental design by (essentially) using the normal covariance matrix $\Sigma$ as a decision variable and the upper bound on the mean squared error as an objective.
Finally, simulations are run to investigate the performance of the method under various types of model-mispecification.

**Strengths:**

The main strength of the paper is a novel method for designing randomized experiments under network interference. The novelty of the method comes from developing an objective function whose decision variables are continuous, (essentially) representing the covariance matrix of assignments. This stands in contrast to previous methods which focus on selecting clusters for independent cluster randomization designs. An additional strength of the paper is that the objective considers both bias and variance, which is uncommon: most approaches attempt to "fix" one of these two things, but typically not both.

One of the most exciting technical connections is the use of Grothendieck's identity, which (to the best of my knowledge) has not been used in causal inference. This offers a new tool in the design of experiments, which I suspect will be useful beyond the specific design used in this paper.

Moreover, the simulations are very well formulated and executed.
In particular, the authors investigate the effectiveness of their design under various forms of model misspecification, which speaks to the robustness of the design.
This is important to investigate via simulations, as formal theory seems difficult given the black-box nature of the experimental design.

**Weaknesses:**

There are several weaknesses of the current method.

1. **Assumed Knowledge**: The paper makes some strong assumptions as to what the experimenter knows. For example, authors assume that the coefficients $\alpha_i$ are known by the experimenter. I think most experimenters will find that knowledge of each individual $\alpha_i$ is too strong of an assumption to be practical -- in the SUTVA setting, this implies that all individual treatment effects can be estimated perfectly without any randomization. In order to be transparent, authors should state this assumption and estimator earlier in the paper, perhaps replacing the HT estimator in (eq 5).
2. **Pre-specified Clusters**: The paper assumes that clusters are pre-specified and little advice is given to practicioners on how to select the clusters in order to minimize MSE.
3. **Understanding of the Variance**: In order to run power calculations, experimenters should have at least a rough understanding (e.g. asymptotic rates) of how the variance of the experimental design depends on sample size $n$ and network parameters. Because this method is based on a black-box optimization procedure, it seems hard to analyze the variance (i.e. optimal value).
4. **Confidence Intervals**: In practice, experimenters value interval estimators (i.e. confidence intervals) more than point estimators, as they provide a method for uncertainty quantification. The necessary tools for uncertainty quantification (e.g. Central Limit Theorem, variance estimation) are not presented in this paper.

While authors should make certain minor changes to address the above, I think that these weaknesses actually constitute further research directions on this exciting method.
Overall, it is my opinion that the strengths of the paper outweigh the weaknesses.

In the two sections below, I discuss minor weakenesses in the technical discussions and literature review which should be addressed by authors before publication.

## Technical Remarks

Below are some minor remarks on technical aspects of the paper.
I think there are a few technical issues in the discussions, but I believe these can be easily fixed and will strengthen the technical contribution of the paper.

1. (Line 137) authors write "without loss of generaltiy, we consider the balanced cluster-level randomization scheme satisfying...$E[z_i] = 1/2$." I think that this restriction is perfectly fine, but I would say it is not technically correct to describe it as "without loss of generality". Consider the usual SUTVA setting: if the outcomes under treatment have more variation than outcomes under control, then the Horvitz--Thompson estimator can be made to have smaller variance by setting $p$ so that treatment is assigned more frequently. I think the phrase "without loss of generality" is not warranted and a simple fix is to just remove it.
2. (Line 152) The word "overparametrized" is not quite standard in this literature so I'd recommend either informally defining it, or just saying that "there are more unknown potential outcomes than observations". This is true even under SUTVA.
3. (Line 208): Authors introduce what they call "Assumption 2". I would refer to this as a "condition" rather than an assumption. The reason is that Assumption 2 only plays a role in choosing the design -- using standard techniques (CLT + variance estimator), Assumption 2 would not be necessary for, say, the validity of confidence intervals. Some readers might misinterpret the use of the word "Assumption" to think "if this condition does not hold, then the estimates are no longer statistically valid in some sense".
4. (Line 254) Authors write "This lemma enables us to sample from bivariates Bernoulli distribution with mean (1/2, 1/2) and any valid covariance". While this is true for $n=2$ variables, I do not believe it to be true for general $n$ variables. It is sort of implied in the Section that this "Grotendeick mapping" can recover any covariance matrix of $\pm 1$ variables. If authors have a proof of this, they should provide it; otherwise, they should clearly state that this "Groethendeick mapping" cannot generate all $\pm 1$ covariance matrices. I think that clarifying this will increase understanding and appreciation of the method.
5. (Line 294): The Monte Carlo simulation is only performed 200 times. In my experience, this is quite low. Can you increase to 10,000 Monte Carlo runs before camera ready submission? This will increase the reader's confidence in you results.

## Literature Review + References

The authors have missed a few important references in the causal inference literature.
I believe these should be easily fixable and would increase the relevancy of the paper by better situating it within the causal inference literature.

1. (Line 36-37) In reference to cluster designs, authors write: "This technique is originally developed in [31] and becomes a prevalent paradigm for network experiment design". I completely agree that [31] was an influential paper for introducing cluster designs to the computer science community. [35] works the exposure mapping framework for interference [1] which allows for this arbitrary network interfernece. However, the so-called "partial interference" assumption has been used since at least Hudgens & Halloran (2008) and cluster designs were advocated for here. So, I'd at least reference some of this early work on clustering in the context of partial interference. I think this will also tie your contributions back to the vien of causal inference in the statistics literature in a stronger way.
2. (Line 38) Authors write that "sharing same treatment within cluster is usually necessary for characterizing the GATE". I would remove or substantially weaken this statement. The proliferation of cluster designs is *not* because they are necessary; but rather, a conceptually simple yet effective type of experimental design.
3. (Line 64) Authors write "In this paper, we propose to treat the covariance matrix of treatment vector as a decision variable in optimization". The following paper seems especially relevant and authors should draw a comparison: Harshaw et al (2019) "Balancing Covariates in Randomized Experiments using the Gram--Schmdit Walk Design". This paper studies experimental designs that directly control the covariance matrix Cov(z) (via discrepancy theory) to bound the variance of Horvitz--Thompson estimator by an implicit ridge regression of outcomes on covariates. A key idea in that paper is to use the operator norm as a measure of worst-case variance, which seems like an alternative to your Assumption 2. Given the similarity in the spirit of the two papers, this paper would benefit from a brief comparison discussion.
4. (Line 94) Authors cite several papers on bipartite experiments. It seems that [15] and [16] are duplicates. To the best of my knowledge, the paper of Zigler & Papadogeorgou (2021) was the first paper to propose bipartite experiments (it has been a working paper since 2019), so a citation is warranted in that discussion.
5. (Line 104): Authors write "Along the same direction, [32] tries to provide..." I recommend revising this language. The word "tries" gives the indication that "[32] tries and fails".
6. (Line 108): In the discussion of partial interference, early work like Hudgens and Halloran (2008) is missing.
7. (Line 116): A citation of several papers with more general forms of interference is listed. The recent paper Harshaw, Sävje, Wang (2022) "A design-based riesz representation framework for randomized experiments" is worth citing, as it proposes a deisgn-based framework which captures and extends previous types of interference.
8. (Line 251): I haven't read [19], but it was my understanding that Grothendieck's identity is typically a different type of statement, where there are two fixed vectors $x$ and $y$ and the random variables are $\textrm{sign}(\langle z , x \rangle)$ and $\textrm{sign}(\langle z , y \rangle)$, where $z$ is uniform from the $\ell_2$ ball. In fact, I have only seen Lemma 1 in certain course notes on Sums-of-Squares (though I'm sure it's appeared in other places). If Lemma 1 does not directly appear in [19] then authors should cite a relevant paper that derives it. In fact, this would probably be helpful to tie your work back to theoretical computer science's use of the technique.

## References

- Harshaw, C., Sävje, F., Spielman, D., & Zhang, P. (2019). "Balancing covariates in randomized experiments with the Gram-Schmidt Walk design". (arXiv:1911.03071)
- Harshaw, C., Sävje, F., & Wang, Y. (2022). "A design-based riesz representation framework for randomized experiments". (arXiv:2210.08698)
- Hudgens, M. G., & Halloran, M. E. (2008). "Toward causal inference with interference". Journal of the American Statistical Association, 103(482), 832–842.
- Zigler, C. M. and Papadogeorgou, G. (2021). "Bipartite causal inference with interference". Statist. Sci., 36(1):109–123.

**Questions:**

1. One way to avoid "picking clusters" is to run the method when each unit is its own singleton cluster. How does the proposed method perform in this case? In other words, is the specification of clusters necessary to achieve small variance? If so, can you describe why or generally comment more on this? (this is a discussion I would find really fascinating in the paper)
2. I claimed that the "Grothendeick mapping" cannot recover all covariance matrices on $\pm 1$ matrices. Can you comment on this?
3. Can you increase the Monte Carlo samples to 10,000?
4. Can you comment on the comparison to the Gram--Schmidt Walk Design of Harshaw et al (2019)?


**Limitations:**

yes

---

> ### Author Rebuttal · Authors · 2023-08-05
>
> Thanks a lot for such a detailed and informative review, and your feedback will undoubtedly contribute to the enhancement of our work. We address your concerns as follows.
>
> ## Confidence interval
>
> We think it poses a significant challenge due to the estimation of nuisance parameters, such as $\gamma$ and $\beta_i$, within the potential outcome model. These parameters play a crucial role in both bias and variance, and the lack of precise information for their estimation necessitates the introduction of additional assumptions when constructing a confidence interval for the GATE estimator. For instance, constructing a conservative confidence interval with variance bound and $\hat\gamma$ may be feasible.
>
> ## Responses to remarks
>
> We are deeply grateful for the technical and literature remarks, which will substantially facilitate our paper's improvement. The following comments are provided in response to some remarks:
>
> For technical remark 5, we can increase Monte Carlo samples to 10,000 before camera-ready submission.
>
> For literature remark 8, the form of lemma 1 in the main body is presented for its intuitive role within our pipeline. We can understand multivariate Gaussian variables $X$ and $Y$ as $\langle g,x\rangle$ and $\langle g,y\rangle$, where $g\sim \mathcal{N}(0,I_n)$ and $x,y$ are two real-valued vectors. This aligns with the original form of Grothendieck identity and is exactly the form employed in our proof of this lemma. In addition, we'll adjust the citation for this identity to a more contemporary paper, wherein it's explicitly presented.
>
> ## Singleton cluster
>
> We think cluster structure isn't necessary in pursuit of variance reduction, since we have a much larger decision space without this pre-specified scheme, and our method can extend to this case naturally. However, an important component of our methodology, $C$ matrix, would be uninformative, and assumption 2 would be vulnerable, in this case.
>
> On the other hand, we maintain that the point lies in estimation error instead of approximation error, and we think the restriction of decision space granted by cluster structures is effective for us to find such a relatively good solution. Empirically, we find that extremely high resolution (>1000, near to singleton cluster case) in clustering corresponds to an increased MSE, which is approximately 20% higher as compared to a resolution of 10.
>
> If scalability is also taken into consideration, then the cluster structures is indispensable.
>
> ## Grothendeick mapping
>
> This is a very interesting question, and we think the answer to it is exactly no. The validity of $\sin(2\pi X)$ as a PSD matrix cannot be guaranteed, even when $X$ functions as a legitimate covariance matrix for a multivariate Bernoulli distribution. This observation is elaborated upon in lines 255 to 257. Moreover, if the recovery procedure is omnipotent, then we'll get the conclusion that the distribution of $n$-varaible Bernoulli vector is uniquely determined, given its mean vector and covariance matrix, since the Gaussian distribution behind is uniquely determined. Paradoxically, mean vector and covariance matrix can't uniquely determine the multivariate Bernoulli distribution for $n>2$ case.
>
> However, this answer bears no impact on our procedures. The parameterization method and the projection step in our algorithm collectively provide a guarantee for sampling from the corresponding Gaussian distribution and the transformed variables are subject to such an optimized covariance matrix, though at a price that we don't search in all possible covariance matrices.
>
> ## Comment on comparison to Harshaw et al. (2019)
>
> We appreciate the significance of the paper by Harshaw et al. (2019) concerning our work. In our forthcoming paper revision, we are committed to incorporating a comprehensive discussion that addresses this connection. The crux of similarity resides in the methodology of optimizing an upper bound of MSE of HT estimator. Moreover, both of them utilize the covariance matrix as decision variable in optimization.
>
> The difference in problem setting, optimization formulation and algorithm are readily discernible, and we primarily comment on the difference in the derivation of MSE bound. We consider interference conducted through social network, where the interference is modeled as an additive term $\gamma \sum_{j \in N_i} z_j$ in our potential outcome model. This component makes covariance appear in the bias of estimator, and there exist second, third, and fourth-order terms w.r.t. treatment vector $t$ in the expression of variance of estimator.  In Harshaw et al (2019), Lemma 3.2 tells us that the influence of experiment design is absolutely captured by $\operatorname{Cov}[z]$, while in our case, the influence of experiment design is much more complex. We derive a variance bound that only contains $\operatorname{Cov}[z]$ through careful analysis, in which we focus on guaranteeing the bound isn't too loose but still optimizable w.r.t. $\operatorname{Cov}[z]$. The techniques and ideas to tackle with higher order term (>2) only appear in our paper, and the situation that experiment design is absolutely captured by $\operatorname{Cov}[z]$ is far from granted for free in our case.
>
> In lines 340~343 of our paper, we provide a discussion about an alternative upper bound of second-order term, namely, the spectral norm of $E[tt^T]$ (also, operator norm, with $l$2-norm), under a weaker comparability assumption. This is exactly the technique applied by Harshaw et al (2019) to bound MSE, while the point of view is different, here we refer to this as "weaker comparability assumption", and it's understood as "worst-case MSE" in Harshaw et al (2019). The logic behind this bound is more natural than that of assumption 2, while this worst-case bound may be too conservative, as evidenced by the performance of IBR in simulation studies, for which similar worst-case variance is utilized.

---

> > ### Comment · Reviewer_1xr3 · 2023-08-11
> > **response to authors**
> >
> > I thank the authors for their thoughtful response to my review.
> > I think many of my questions were answered appropriately.
> >
> > I want to just emphasize a few aspects of my review:
> >
> > - It seems that we are both in agreement that the Grothendeick mapping cannot generate all +/- 1 covariance matrices. All I recommend here is that the language is updated so that readers do not get this impression.
> > - I think your paper makes a good contribution without the development of confidence intervals. I raised this point only as a weakness, not one that I think needs to be addressed in this paper.

---

> > > ### Author Response · Authors · 2023-08-13
> > >
> > > Thank you once more for your dedicated efforts in providing feedback on our paper. We will certainly add comments after this identity to enhance the clarity. Additionally, the issues remarked as weaknesses will be discussed in the future directions of the upcoming version.

---

### Official Review · Reviewer_Q5Zi · 2023-07-06

**Soundness:** 3 good
**Presentation:** 3 good
**Contribution:** 3 good
**Rating:** 7
**Confidence:** 3

**Summary:**

In this paper, the authors propose a new algorithm for A/B test design under network effect with cluster level randomization. By derivation of an upper MSE upper bound with bias-variance trade-off, the authors reparameterize to directly optimize the covariance of the treatment vector. An efficient PGD algorithm is proposed for efficient optimization. Simulation study on a real-world network demonstrates the effectiveness of the proposed method.

**Strengths:**

1. The authors study a very important and practical problem in A/B test design with reasonable assumptions on cluster level randomization.
2. The paper is well written and the proposed method is both explained clearly, with reasonable assumptions  and well justified with detailed derivations.
3. The proposed reparameterization and the PGD optimization algorithm makes  the algorithm practical and easy to implement.


**Weaknesses:**

1. It seems that the clustering algorithm plays a very important role in the quality of the A/B experiment design. From the covariate matrix formulation, maybe in some cases it would be better to directly merge two clusters instead of using the treatment vector to guarantee the same treatment. It would be very interesting to see how the proposed method performs under different cluster methods /parameters besides the number of clusters.
2. As A/B experiments are usually used in real large-scale networks, it would be interesting to see scalability results in terms of number of clusters for different methods under comparison.
3. As the authors point out in the future work, it would also be interesting to see some empirical evaluation on the tightness of the proposed upper bound on the MSE. It may provide some insight on whether further tightening the bound can bring additional benefits.


**Questions:**

Please see above

**Limitations:**

Yes

---

> ### Author Rebuttal · Authors · 2023-08-05
>
> First of all, thanks for your time and efforts in the reviewing process and we appreciate your approval of the theoretical and empirical contributions of this work. We provide some comments on your interesting questions as follows.
>
> ## Clustering issue
>
> The role of clustering indeed introduces intriguing complexities within the sphere of cluster-level design. We once contemplated an end-to-end approach that seamlessly integrated upstream clustering into the design pipeline. However, the intricate interplay between algorithm settings and the resultant clustering structure—particularly when dealing with heuristics-rich community detection algorithms like Louvain—posed formidable analytical challenges. Consequently, we opted to focus on presenting numerical outcomes achieved by varying the resolution parameters of the Louvain algorithm. Notably, the resolution stands as a paramount parameter within our context, aligning with the preferences of our industrial collaborator. It's worth mentioning that a more nuanced analysis might become feasible should cluster information be incorporated in a simple format within the experiment design. An example is the independent block randomization (IBR) that retains cluster information exclusively as cluster sizes [4], which could potentially facilitate an encompassing analysis inclusive of clustering step. As contrast, the cluster structure is involved in our derived bias and variance with matrix $C$, which characterizes the connections within/between clusters.
>
> Furthermore, the prospect of cluster mergers raises pertinent considerations. In this context, we find the optimized covariance matrix to be particularly enlightening. A correlation nearing 1 (equivalent to covariance near 1/4) signifies a distinct entwinement between clusters, akin to a merger from the perspective of treatment assignment.
>
> ## Scalability
>
> Our formulation's key scalability advantage is rooted in its continuous optimization structure, while explicit time complexity analysis proves challenging. Therefore, we provide some comments here. CR and Ber are the most efficient and scalable methods disregarding their performance. IBR builds on dynamic programming that partitions clusters into communities and is relatively scalable. The PSR and ReAR are competitive in performance, while PSR relies on a sequential assignment process and is thus poor on scalability. ReAR relies on PSR and rerandomization, which is difficult to implement in the industrial scene.
>
> ## Empirical evaluation on bound tightness
>
> We appreciate the valuable suggestion to delve into the empirical evaluation of bound tightness. In response, we highlight two pivotal tradeoffs that inherently shape the intricacies of our analysis. The first tradeoff surfaces between the underlying assumption and the tightness of the bound. We illustrate this interplay in lines 340 to 343 by introducing an alternative bound under a weaker comparability assumption. While this bound appears naturally, empirical evidence underscores its excessive conservatism, thereby illuminating the challenge of striking the delicate balance between assumption strength and bound tightness. The second tradeoff stems from the equilibrium between optimization complexity and bound tightness, as meticulously discussed in lines 334 to 339.
>
> Moreover, we acknowledge the significance of equation 19 in the appendix—an integral component of our methodology. This step, $\operatorname{trace}\left(\mathbb{E}\left[\boldsymbol{C} t t^T \boldsymbol{C} t t^T\right]\right) \leq \operatorname{trace}\left(\mathbb{E}\left[\boldsymbol{C} \mathbf{1 1}^T \boldsymbol{C} t t^T\right]\right)$, intrinsically linked to the randomization scheme behind treatment vector $t$, is variably influenced by different experiment designs. This inherent variability poses challenges in deriving a uniform conclusion, thereby rendering the attainment of a definitive outcome a complex endeavor. Nevertheless, it's worthwhile for future extension.

---

> > ### Comment · Reviewer_Q5Zi · 2023-08-13
> > **Thank you for the rebuttal**
> >
> > Thanks for the response. I have read the rebuttal and remain my score.

---

> > > ### Author Response · Authors · 2023-08-13
> > >
> > > Thank you once more for your decision and efforts in the review process, and we'll supplement discussion on your questions in the upcoming version.

---

### Official Review · Reviewer_2fwK · 2023-07-24

**Soundness:** 3 good
**Presentation:** 3 good
**Contribution:** 2 fair
**Rating:** 5
**Confidence:** 3

**Summary:**

This paper presents an optimized covariance design for A/B tests on social networks with interference. The authors address the challenge of accurately estimating the global average treatment effect (GATE) in the presence of network interference. They propose a method to balance bias and variance in experimental design by optimizing the covariance matrix of the treatment assignment vector. The paper derives the bias and variance of the estimator and proposes an algorithm to implement the desired randomization scheme. Simulation studies demonstrate the advantages of their method over existing methods in various settings. Overall, the paper contributes an approach to improve the estimation of treatment effects in social network experiments by optimizing the covariance design.

**Strengths:**

This paper presents a approach to optimize the covariance design for A/B tests on social networks with interference, which improves the estimation of treatment effects and enables more accurate decision making based on experimental results. The paper is well-written and clearly explains the problem formulation, assumptions, and the proposed method. The authors provide a thorough analysis of the bias and variance of the estimator, a upper bound for the MSE of the HT estimator, and a projected gradient descent algorithm to solve the optimization problem. The simulation studies demonstrate the effectiveness and robustness of the proposed method in various settings.

**Weaknesses:**

1.	The paper briefly mentions assumptions about the direct treatment effect and interference effect. However, a more thorough discussion on the validity and generalizability of these assumptions would be valuable. Additionally, discussing the limitations of the proposed method, such as its sensitivity to certain parameter values or potential biases introduced by the assumptions, would provide a more comprehensive understanding of the method's applicability and potential drawbacks.
2.	In the experimental results presented in Tables 2, 5, 6, 7, 10, 11, 14, 15, 19, 22, and 23 in the appendix, it is observed that the proposed algorithm’s performance in terms of MSE is significantly weaker compared to other benchmark algorithms such as ReAR, IBR-p, and PSR. This suggests that further improvements may be necessary for the proposed algorithm to achieve competitive performance
3.	The paper could benefit from a more detailed introduction and explanation of the motivation behind balancing bias and variance. Providing additional context and background information on this topic would help to strengthen the overall argument and improve the clarity of the paper.


**Questions:**

Please refer to weaknesses 1, 2, and 3

**Limitations:**

While the paper briefly mentions assumptions about the direct treatment effect and interference effect, a more thorough discussion on the validity and generalizability of these assumptions would be valuable. Additionally, it would be beneficial for the authors to include a discussion on the limitations and potential negative consequences of their work. This could include an examination of the proposed method’s sensitivity to certain parameter values or potential biases introduced by the assumptions. Providing a more comprehensive understanding of the method’s applicability and potential drawbacks would strengthen the overall manuscript

---

> ### Author Rebuttal · Authors · 2023-08-05
>
> First of all, thanks for your time and efforts in the reviewing process, and we address your concerns as follows.
>
> ## Comparability assumption
>
> The comparability assumption is predicated on the premise that the magnitude of interference remains akin to the direct effect at the cluster level. This assertion inherently excludes scenarios where interference is significantly weaker compared to the direct effect—a stance that aligns with our targeted investigative scope. It's worth noting that our primary focus is on experiment design encompassing network interference. Consequently, this assumption finds its rational grounding in this context. Assumptions on the form and intensity of interference are very common in related works.
>
> In the second place, as the degree of tolerance on incomparability between direct causal effect and interference, $\omega$, escalates, the constraint imposed by this assumption gradually relaxes. We posit that $\omega$ can be perceived as a tunable parameter, encapsulating the practitioner's belief in the degree of comparability. Importantly, the parameter $\omega$ signifies the extent to which a unit's outcomes are contingent upon the activities of its associates, each with varying degrees of influence—a manifestation of real-world complexity. This inherent variability mirrors the natural heterogeneity observed in real-world scenarios. We also recognize and discuss the sensitivity of our results to $\omega$ in line 219 due to limited space. Moreover, the simulation section delves into the sensitivity analysis of other parameters within the potential outcome model, further reinforcing the comprehensiveness of our investigation.
>
>
> ## Clarification on non-competitive performance
>
> First and foremost, our randomization scheme OCD is tailored to the network experiment with interference comparable to direct causal effect. As such, we think it's acceptable that our scheme exhibit relatively diminished performance in scenarios with low interference levels ($\gamma$ set to 0.5). Notably, our method consistently excels as interference intensifies—a pattern that aligns with our intended scope and objectives.
>
> Secondly, it's imperative to contextualize our simulation settings. Both potential outcome models employed mirror instances of model misspecification, presenting considerable challenges. Furthermore, our chosen settings inadvertently favor certain methods—such as considering node degree explicitly as a covariate in both PSR and ReAR, whereas this is  embedded in our potential outcome models in simulation.  Despite these challenges, our method consistently performs commendably across a multitude of settings.
>
> It's worth highlighting that expecting a single experiment design to outperform all alternatives across diverse scenarios is a lofty anticipation. Each randomization scheme inherently thrives within specific contexts. For instance, when γ approaches zero, complete randomization (CR) can potentially lead across all methods.
>
> Lastly, while performance enhancement is indeed a noteworthy facet of our contributions, it's pivotal to recognize that our method's significance extends beyond this. In practical terms, our approach demonstrates commendable efficiency and scalability in the domain of social networks on online platforms. From a theoretical perspective, the meticulous formulation of the problem and the creativity with which the optimization problem is constructed holds paramount importance.
>
>
> ## Motivation for balancing bias-variance
>
> As underscored in our introduction, a predominant focus within existing literature centers on variance reduction. However, we advocate for a paradigm shift, asserting that Mean Squared Error (MSE) serves as a more apt optimization objective. This pivot inherently embodies the equilibrium between bias and variance—a fundamental tenet of the bias-variance tradeoff.
>
> Delving into Section 2.3, we undertake a comprehensive analysis of the sources underpinning bias. Within the realm of social networks, bias can manifest substantially, serving as a pivotal driver for our decision to holistically address both bias and variance. This awareness underscores our commitment to embracing a balanced perspective, effectively navigating the intricate interplay between these critical elements. More intuitively, when both bias and variance are considerable, then we should not expect an unbiased estimator or constant estimator to be optimal.

---

> > ### Comment · Reviewer_2fwK · 2023-08-11
> >
> > I appreciate your thorough response, which addressed several of my concerns. As a result, I have decided to revise my rating upwards.

---

> > > ### Author Response · Authors · 2023-08-13
> > >
> > > Thank you once more for your decision and the time you've invested in reviewing our paper, and we will supplement discussion on your concerns in the upcoming version for enhancing the positioning of this paper.

---

### Decision · Program_Chairs · 2023-09-21

**Decision:**

Accept (poster)

**Comment:**

The paper studies the experimental design for A/B tests in social networks, which may have interference due to individuals in social networks influencing one another. The authors propose a method to balance bias and variance and design an algorithm to achieve a better performance. The reviewers provided concrete comments and critics, for which the authors gave detailed replies. The reviewers are satisfied by the authors' rebuttal, and two of them raised their scores, indicating that the authors' rebuttals are effective, and the reviewers are confident in supporting the paper after the authors' explanations. The topic is an important one, and the study provides a nice addition to the research community on this topic. I recommend acceptance to the paper.